# Reasoning LLM Improves Speaker Recognition in Long-form TV Dramas

Yuxuan Li [* 1]  Lingxi Xie [* 2]  Xinyue Huo [2]  Jihao Qiu [3]  Jiacheng Shao [2]  Pengfei Chen [2]  Jiannan Ge [2]
Kaiwen Duan [2]  Qi Tian [2 4]

## Abstract

Long-form TV dramas present a formidable challenge for comprehensive video understanding, where deciphering complex storyline often relies on **speaker recognition**, the task of accurately attributing each spoken utterance to its respective character. In this paper, we advance this field through two primary contributions. (1) We introduce **DramaSR-532K**, a large-scale benchmark comprising 532K annotated dialogue lines across more than 900 unique characters, necessitating the integration of auditory, linguistic, and visual cues for speaker recognition. (2) We propose **DramaSR-LRM**, a robust approach built upon a large reasoning model (LRM). DramaSR-LRM is designed to autonomously aggregate contextual evidence via multimodal tool-use, synthesizing diverse inputs to achieve high-fidelity attribution. Experimental results demonstrate that DramaSR-LRM significantly outperforms existing baselines, particularly on short utterances where acoustic biometrics are inherently unreliable. *All the data and code will be made publicly available at the project page:* `https://www.github.com/198808xc/DramaSR-LRM`.

## 1. Introduction

In recent years, the rapid evolution of multimodal large language models (MLLMs) has significantly advanced long-form video understanding. However, deciphering long-form TV dramas remains a formidable challenge, particularly regarding storyline that hinge on the complex interactions and evolving relationships of a large cast. A foundational requirement for this task is speaker recognition (SR), the subtask of attributing each utterance to its respective character. While a vast corpus of research exists for speaker diarization and verification, existing literature rarely addresses the unique complexities of TV dramas. In such settings, an intelligent agent must synthesize disparate multimodal cues, including visual frames, acoustic signals, and linguistic context, to achieve accurate attribution.

To address this gap, we introduce **DramaSR-532K**, a large-scale benchmark curated from 13 long-form TV series, some of which span multiple seasons. Within DramaSR-532K, SR is formulated as an open-set classification problem: characters are defined either by proper names for primary roles (*e.g.*, *Alice*, *Bob*) or by descriptive attributes for secondary roles (*e.g.*, *Man in Black*, *Female Doctor*). Our annotation pipeline consists of two stages. First, we extracted hard-rendered subtitles to define utterance boundaries and computed corresponding acoustic biometrics (voiceprints). Leveraging face recognition results, we employed data clustering and affinity propagation to generate initial pseudo-labels. Second, we conducted a rigorous human-in-the-loop revision process to refine these labels and resolve ambiguous cases. The resulting dataset comprises over 532K utterances across more than 900 primary and more than 6.6K secondary roles, establishing DramaSR-532K as the most comprehensive benchmark to date for speaker recognition in movies or TV dramas (see Table 1).

To provide a robust solution for this benchmark, we propose DramaSR-LRM, a Large Reasoning Model (LRM) architected to utilize three specialized tools: (a) `voice_sim`: an acoustic utility that computes cosine similarity between the target voiceprints and known character embeddings; (b) `video_cap`: a video captioning module that extracts visual context of the utterance; (c) `char_relation`: a relational database that assists parsing social hierarchies and appellations (*e.g.*, 'Mom' and 'Darling') within the dialogue. To train DramaSR-LRM, we leveraged a single drama comprising 50K utterances, and utilized Gemini-3-Pro (Team et al., 2023) to generate high-quality supervised fine-tuning (SFT) trajectories. We fine-tuned a Qwen3-8B backbone (Yang et al., 2025) using a pipeline of SFT followed by reinforcement learning (RL) to optimize reasoning consistency.

Expect for 2 dramas leveraged for training process, eval-

---

*Equal contribution  [1]Tsinghua University, China  [2]Huawei Inc., China  [3]University of Chinese Academy of Sciences, China  [4]Guangdong Laboratory of Artificial Intelligence and Digital Enonomy (SZ), China.  Correspondence to: Lingxi Xie <198808xc@gmail.com>, Qi Tian <tian.qi1@huawei.com>.

| Benchmark | source | # videos | tot. dura. | avg. dura. | # speakers[†] | # uttr. | language |
|---|---|---|---|---|---|---|---|
| AMI (Carletta et al., 2005) | meeting | 684 | ≈100h | 8.8mins | 3 / 4.0 / 5 | – | En |
| VoxConverse (Chung et al., 2020) | TV show | 448 | 63h,50mins | 8.5mins | 1 / 5.6 / 21 | – | En |
| AVA-AVD (Xu et al., 2022) | movie | 351 | 29h,15mins | 5.0mins | 2 / 7.7 / 24 | – | Multi |
| MSDWild (Liu et al., 2022) | V-log | 3,143 | 80h,03mins | 1.5mins | 2 / 2.7 / 10 | 112K | Multi |
| **DramaSR-532K (ours)** | TV drama | 806 | **525h,31mins** | **39.1mins** | **30 / 69.5 / 144** | **532K** | En&Cn |

*Table 1.* DramaSR-532K is different from existing benchmarks for audio-video speaker diarization in terms of (total and average) video duration, and number of speakers (min/average/max per drama; [†]only known characters are counted), and number of utterances.

uation across the remaining 11 dramas demonstrates that DramaSR-LRM elevates the accuracy of the label propagation baseline from 85.49% to 87.79% (a 2.30% gain). These gains are particularly pronounced in hard scenarios, in particular, short (a 3.33% gain) and very short (a 9.20% gain) utterances, as well as dramas with a relatively low baseline (*e.g.*, *Lost*: a 5.16% gain, and *Qin Empire 2*: a 4.06% gain). Our work underscores the critical role of speaker recognition in long-form video understanding and provides a scalable framework for future research, which can potentially address more challenges in our benchmark, *e.g.*, **open-world** speaker description and **end-to-end** speaker recognition.

## 2. Related Work

**Speaker Recognition** aims to identify a speaker's identity through vocal utterances (Bai & Zhang, 2021; Kabir et al., 2021), serving as a fundamental component in diverse audiovisual understanding pipelines. Our study is highly related to three established research domains.

**(a) Speaker Diarization** (SD) involves partitioning an audio or video stream into homogeneous segments attributed to specific individuals (*i.e.*, 'who spoke when') (Park et al., 2022). While traditional SD focused primarily on audio-only inputs (Carletta et al., 2005), the field has transitioned toward multimodal frameworks (Liu et al., 2022; Xu et al., 2022) and joint diarization-recognition models in the era of deep learning (Cornell et al., 2024; Efstathiadis et al., 2025; Yin et al., 2025). However, significant gaps remain. First, existing SD benchmarks typically involve a limited number of speakers (*e.g.*, 10–25 in AVA-AVD (Xu et al., 2022)), whereas a single TV drama may feature over 100 named characters alongside numerous transient roles. Second, since our work leverages timestamped dialogue transcriptions, the challenge shifts from temporal segmentation to character attribution in complex narrative contexts.

**(b) Speaker Verification** (SV) requires determining whether a given utterance matches a target identity (Mittal & Dua, 2022; Tu et al., 2022). Prominent benchmarks like VoxCeleb (Nagraniy et al., 2017; Chung et al., 2018) and CN-Celeb (Fan et al., 2020) have scaled this task using millions of web-sourced utterances, while others like 3D-Speaker (Zheng et al., 2023) explored cross-device and cross-dialect robustness. While acoustic features derived

from state-of-the-art SV models (Wang et al., 2023; Chen et al., 2024) provide a strong foundation for our work, TV dramas introduce unique difficulties: multiple characters often interact within the same acoustic environment, leading to overlapping speech and varying recording conditions that go beyond standard verification settings.

**(c) Active Speaker Detection** (ASD), in the visual domain, focuses on identifying which visible characters are currently speaking by correlating facial movements with audio signals. Established datasets such as Columbia (Chakravarty et al., 2016) and AVA-ActiveSpeaker (Roth et al., 2020) have driven the development of unified audio-visual architectures (Alcázar et al., 2021; Liao et al., 2023) and temporal modeling techniques (Tao et al., 2021; Min et al., 2022). While ASD identifies the presence of a speaker in a frame, our task, in the context of long-form dramas, extends it by maintaining character identity across multiple scenes and seasons, even when the speaker is off-screen.

**In summary**, these speaker recognition paradigms are mutually reinforcing (Wang et al., 2024b) and provide critical metadata for downstream applications, including dense video captioning (He et al., 2024), speech-driven video generation (Zhi et al., 2023), and semantic video editing (Wang et al., 2025).

**Long-form video understanding** has gained significant traction within the research community. While several benchmarks for hour-long video QA have recently emerged (*e.g.*, EgoSchema (Mangalam et al., 2023), LongVideoBench (Wu et al., 2024), and Video-MME (Fu et al., 2025)), efforts toward long-form TV dramas remain relatively scarce. In narrative-driven media, speaker recognition is a pivotal task, as establishing 'who speaks what' is essential for deciphering intricate storylines and character dynamics. Our study includes preliminary evaluations demonstrating that robust speaker attribution is a critical prerequisite for holistic drama understanding.

**Large language models** (LLMs) (Brown et al., 2020; Touvron et al., 2023; Team et al., 2023) have demonstrated versatile capabilities across a wide range of AI applications. Two critical evolutions of this technology are particularly relevant to our work: multimodal LLMs (MLLMs) (Liu et al., 2023; Zhu et al., 2023; Achiam et al., 2023; Team et al., 2024; Bai et al., 2025b), which integrate visual percep-

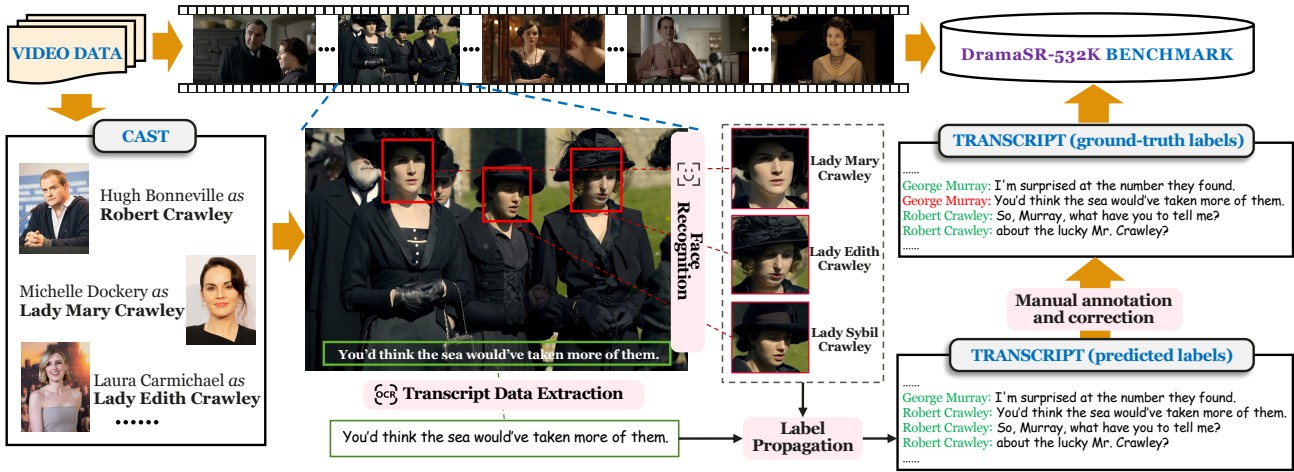

*Figure 1.* How the DramaSR-532K benchmark was established. We extract (1) transcript data from OCR, (2) cast information from ending credits and web data, and (3) perform label propagation followed by human annotation to obtain the ground-truth labels.

tion with linguistic reasoning, and large reasoning models (LRMs) (Guo et al., 2025; Comanici et al., 2025; Yang et al., 2025), which utilize extended inference-time processing to solve complex problems. In our framework, an MLLM is employed for fine-grained video clip description, while an LRM is specifically trained to synthesize disparate multimodal cues for speaker recognition. It is related to (Wang et al., 2024a), which applied an LLM for speaker diarization post-processing.

The optimization of our LRM relies on reinforcement learning (RL). We build upon classical algorithms like PPO (Schulman et al., 2017) and more recent advancements such as GRPO (Shao et al., 2024), which enhances policy optimization efficiency. While recent literature has explored RL-based policy refinement for both LLMs (Yu et al., 2025; Zheng et al., 2025; Zhao et al., 2025) and MLLMs (Huang et al., 2025; Zhan et al., 2025), its application to narrative speaker identification remains underexplored. Notably, while SpeakerLM (Yin et al., 2025) and D-ORCA (Tang et al., 2026) utilize MLLMs for speaker identification, our approach is distinct in its use of a reasoning-centric model to explicitly aggregate multimodal evidence in complex scenarios.

## 3. The DramaSR-532K Benchmark

A conceptual illustration of the benchmark, including our data preparation and annotation pipeline, is shown in Figure 1. Formally, processing a long-form TV drama (which typically encompasses dozens of hours of video) requires a multi-stage pipeline: temporal speech segmentation, character discovery (indexing the cast), and ultimately, character attribution. To isolate the challenges of high-level reasoning, we adopt a simplified setting by assuming that: (1) utterances are pre-segmented and synchronized with their

corresponding dialogue transcriptions, and (2) a comprehensive candidate character list is provided *a priori*. Thus, the primary objective is to accurately attribute each utterance to its respective character from the candidate set.

Despite these boundary conditions, the task remains inherently complex. As depicted in Figure 1, several 'corner cases' frequently arise: (a) **short utterances** provide insufficient signal for stable voiceprint extraction, (b) **environmental complexity**: in multi-character scenarios, background noise and overlapping speech introduce significant signal interference, and (c) **visual absence**: the speaking character may be off-screen or occluded, raising difficulties for visual-based identification. These challenges necessitate the development of a sophisticated reasoning model (see Section 4) capable of synthesizing disparate multimodal cues, including character detection (primarily via facial recognition), the alignment of acoustic biometrics (voiceprints), and the logical parsing of dialogue context, to achieve high-fidelity recognition.

### 3.1. Data Preparation

Our benchmark is established upon 13 popular long-form TV dramas (3 English and 10 Chinese productions), encompassing a total duration of 525 hours. We denote each raw video data as V and the total temporal duration as $T$. The data preparation pipeline consists of two primary phases: transcript extraction and character library construction.

**Transcript extraction.** We first deploy an OCR model (*i.e.*, PaddleOCR (Cui et al., 2025b;a)) to extract hard-rendered subtitle text directly from the video stream. To ensure linguistic and temporal precision, we deduplicate the results and utilize an MLLM (*i.e.*, Qwen2.5VL (Bai et al., 2025b)) to refine the transcriptions by analyzing correspond-

ing keyframes. For approximately 1% of the data identified as anomalous, where the ratio of temporal duration to text length deviates significantly from normal rates, we conduct manual verification and correction. This process yielded a final corpus denoted as $\mathcal{S} = \{(\mathbf{t}_n, a_n, b_n)\}_{n=1}^N$, where each $\mathbf{t}_n$ denotes the transcription text of utterance $\mathbf{u}_n$, and $a_n < b_n \in (0, T)$ represent the start and end timestamps, respectively.

**Character library construction.** To define the candidate speaker set, we first extract actor-role mappings from the series' credits using OCR (supplemented by manual verification) and cross-referenced with public web metadata. A character (facial) library is initialized with professional portraits of the identified cast members, and iteratively expanded by scanning the entire drama at 1 FPS to collect diverse, *in-situ* facial samples. Then, we perform frame-wise facial detection and recognition across the dataset. Consequently, we obtain collection of $M$ visual instances $\mathcal{F} = \{(\mathrm{id}_m, c_m)\}_{m=1}^M$, where $\mathrm{id}_m \in \{1, 2, \ldots, P, \ldots, P+U\}$ identifies the character ID from a pool of $P$ identified individuals (where $\mathrm{id} \leqslant P$) and $U$ ancillary (unrecognized or background) individuals, and $c_m$ denotes the timestamp of the facial instance.

**Task formulation.** Formally, the objective of speaker recognition is to produce $\mathcal{R} = \{r_n\}_{n=1}^N$, where $r_n$ is the character assignment for utterance $\mathbf{u}_n$, denoted as $r_n \in \{0, 1, \ldots, P\}$, where $r_n > 0$ corresponds to a specific character ID from the library, and $r_n = 0$ serves as the null class, indicating that the speaker is an ancillary character. For further technical details of data preparation, please refer to Appendix A.1.

### 3.2. Speaker Label Annotation

To establish a high-fidelity ground truth, we conducted a human annotation procedure. To reduce the annotation overhead, we employed a semi-automated pipeline: an initial pass was performed using the label propagation baseline (see Section 4.2), which achieved an estimated accuracy of 90%. Human annotators were then tasked with auditing and refining these pseudo-labels. We developed an graphical user interface that synchronizes a video player with real-time dialogue contexts, acoustic biometric statistics, and candidate character profiles to maximize labeling efficiency.

Beyond standard character attribution, we established protocols for three special taxonomies. (a) When a speaker is not present in the existing character library, labelers infer the character's name from narrative context or assign a descriptive identifier (*e.g.*, *Man in Black*, *Alice's Father*). (b) In instances where the speaker's identity remains ambiguous despite multimodal evidence, a special [UNKNOWN] tag is assigned to maintain dataset integrity. (c) For dialogue lines spoken by multiple characters (*e.g.*, choral speech or rapid-fire interruptions), annotators partition the transcrip-

tion into segments and assign the appropriate speaker(s) to each specific sub-unit. On average, an experienced annotator requires $1.5$ to $2.5$ hours to process one hour of video content. Comprehensive details regarding the quality control measures are provided in Appendix A.3.

We evaluated the inter-annotator agreement in about 10% of data. On average, two annotators agreed on about 99.6% of subtitle lines. By estimation, the final label noise is about 0.5%, far higher than all the algorithms (10+%).

### 3.3. Evaluation Protocol

The efficacy of speaker recognition models is assessed by comparing predicted labels against the manually verified ground truth. For each utterance, we evaluate the prediction accuracy based on the following taxonomic logic:
**(a) Single-speaker utterances.** If the ground truth identifies a specific primary character, the prediction is marked correct only if it matches that character's ID. For utterances attributed to ancillary roles in the ground truth, the prediction is considered correct if it maps to the 'null' class.
**(b) Multi-speaker utterances.** A prediction is deemed correct if it identifies any of the contributing speakers within the ground-truth set or correctly assigns the 'null' label.
**(c) Unknown ground truth.** Any prediction is considered valid to ensure these ambiguous samples do not unfairly penalize model performance.
We will show that the evaluation protocol does not change the conclusion (that our method is better than the baseline) in Section 5.2.

We report utterance-wise prediction accuracy for each drama. To provide a granular understanding of model robustness, we also evaluate performance across several challenging subsets:
**(a) Temporal duration.** Categorizing utterances by length, as brief speech fragments often lack stable acoustic biometrics for recognition.
**(b) Character density.** Categorizing utterances by the number of candidate characters to assess the model's ability to disambiguate in crowded environments.
**(c) Visual occlusion.** Isolating utterances where the speaker is not visible in context, forcing the model to rely exclusively on acoustic and linguistic reasoning.

**Benchmark extensibility.** Beyond the core task described above, the rich annotations in DramaSR-532K support more sophisticated research frontiers. These include open-world speaker identification, where models must generate linguistic descriptions for ancillary characters, and end-to-end speaker recognition, which requires simultaneous speech segmentation, character discovery, and attribution from raw video. These extensions ensure the long-term utility of the benchmark for the evolving field of multimodal AI.

# 4. Methodology

## 4.1. The LRM-based Pipeline

The pipeline of our method is detailed in Algorithm 1. It begins with an initialization phase utilizing a label propagation algorithm, followed by an iterative refinement phase driven by a large reasoning model (LRM). During each iteration, the LRM dynamically invokes a suite of multimodal tools, including voiceprint similarity, video captioning, and character relationship, to adjudicate ambiguous cases and resolve conflicting evidence. These tool outputs are updated periodically to incorporate the refined labels from the preceding round, enabling the model to converge on a high-fidelity narrative understanding. In the following subsections, we first detail the mechanics of the label propagation baseline and the technical implementation of our multimodal toolset. Subsequently, we describe the training paradigm for the LRM, including the integration of supervised fine-tuning and reinforcement learning.

---

**Algorithm 1** Overall Pipeline of **DramaSR-LRM**

---

**Require:** raw video data $\mathsf{V}$, transcript corpus $\mathcal{S}$, facial instance set $\mathcal{F}$
**Ensure:** speaker attribution result $\mathcal{R}$
    Initialization: $\mathcal{R} \leftarrow \texttt{label\_prop}(\mathsf{V}, \mathcal{S}, \mathcal{F})$    $\triangleright$ Sec 4.2
    **while** not converged **do**
        Similarity: $\mathbf{K} \leftarrow \texttt{voice\_sim}(\mathsf{V}, \mathcal{S}, \mathcal{R})$   $\triangleright$ Sec 4.3
        Captioning: $\mathcal{C} \leftarrow \texttt{video\_cap}(\mathsf{V}, \mathcal{S}, \mathcal{R})$   $\triangleright$ Sec 4.3
        Relation: $\mathcal{X} \leftarrow \texttt{char\_relation}(\mathcal{S}, \mathcal{R})$   $\triangleright$ Sec 4.3
        LRM refinement: $\mathcal{R} \leftarrow \texttt{LRM}(\mathcal{R}; \mathcal{P}, \mathbf{K}, \mathcal{X})$ $\triangleright$ Sec 4.4
    **end while**

---

## 4.2. Label Propagation

**Feature extraction.** To initialize speaker attribution, we first extract acoustic embeddings for each dialogue line. Using the start and end timestamps, we segment the raw audio into discrete clips. Each clip is padded with a $100\mathrm{ms}$ buffer at both boundaries; to prevent temporal overlap with adjacent segments, the buffer is truncated at the midpoint between neighboring utterances where necessary. We utilize ERes2Net (Chen et al., 2023), a pre-trained speaker verification model[1], to extract a 192D feature vector $\mathbf{v}_n$ for the $n$-th utterance. These vectors are $\ell_2$-normalized to facilitate cosine similarity measurements within the range of $[-1, 1]$.

**The neighborhood assumption.** Our initialization relies on a spatiotemporal neighborhood assumption: the visual presence of a character suggests a possibility of that character

---

[1]In a nearest-neighbor classification test using the training set, ERes2Net features consistently outperformed other candidates, including ECAPA-TDNN (Desplanques et al., 2020), CAM++ (Wang et al., 2023), and ERes2Net-v2 (Chen et al., 2024).

speaking within a temporal window of $\tau = 30\mathrm{s}$[2]. Formally, for each character $p \in \{1, 2, \ldots, P\}$, we define the facial instance set $\mathcal{F}_p = \{(\mathrm{id}_m, c_m) \in \mathcal{F} \mid \mathrm{id}_m = p\}$. We then identify the candidate utterance set $\mathcal{S}_p$ for character $p$ as $\mathcal{S}_p = \{(\mathbf{v}_n, a_n, b_n) \in \mathcal{S} \mid \exists m, (\mathrm{id}_m, c_m) \in \mathcal{F}_p \wedge a_n - \tau \leqslant c_m \leqslant b_n + \tau\}$.

**Seed cluster generation.** Within each set $\mathcal{S}_p$, dense clusters in the acoustic embedding space typically correspond to the target character. However, to mitigate false attributions, particularly for secondary characters who frequently co-occur with primary characters, we implement a greedy step-wise clustering procedure. In each step, we identify the maximum cluster across all $\{\mathcal{S}_p\}$ and assign it to the respective character. Subsequent clusters exhibiting excessive similarity to previously labeled sets are discarded to maintain purity. This process continues until each known character is associated with at least one seed voiceprint set[3]. The $p$-th character's seed set is denoted as $\mathcal{V}_p$.

**Affinity propagation.** We subsequently apply the neighborhood assumption in reverse: for any given utterance, the candidate speaker pool is restricted to characters whose faces appear within the $\pm\tau$ window[4]. The affinity between an utterance $\mathbf{v}_n$ and character $p$, denoted $k_{n,p}$, is calculated as the mean cosine similarity of the top-$N_p'$ most similar samples in $\mathcal{V}_p$, where $N_p' = |\mathcal{V}_p|^{0.4}$ is an empirically derived scaling factor. We employ an iterative refinement strategy with a monotonically decreasing similarity threshold $\theta$; utterances exceeding $\theta$ are attributed to the matching character and integrated into the respective seed set. The process terminates when $\theta$ reaches a lower bound (*e.g.*, 0.4), at which point any unassigned utterances are labeled as [UNKNOWN].

For further technical details of the label propagation algorithm, please refer to Appendix A.3.

## 4.3. Multimodal Toolset

**Voiceprint similarity.** We compute a similarity matrix $\mathbf{K} \in \mathbb{R}^{N \times P}$, where each entry $k_{n,p}$ represents the simi-

---

[2]While this neighborhood assumption restricts candidate sets for both baseline and evaluation, possibly biasing the task in favor of visually anchored methods and under-representing off-screen speech difficulty, using it to add 'visual anchors' is necessary in the current form because a long-form drama has many (30–100+) characters. As we shall see in experiments, our algorithm can recognize off-screen speakers by first assigning it an [OTHERS] tag and, when necessary, extending the candidate list by adding 3 to 5 characters with the most similar voiceprints, and performing another reasoning step.

[3]In the data annotation stage, to ensure the integrity of these seeds, a rapid human-in-the-loop verification is performed, requiring approximately 1–2 hours per drama.

[4]While this visual-anchor strategy may struggle with off-screen narration or low-visibility scenes, it provides a robust initialization for subsequent refinement by the LRM.

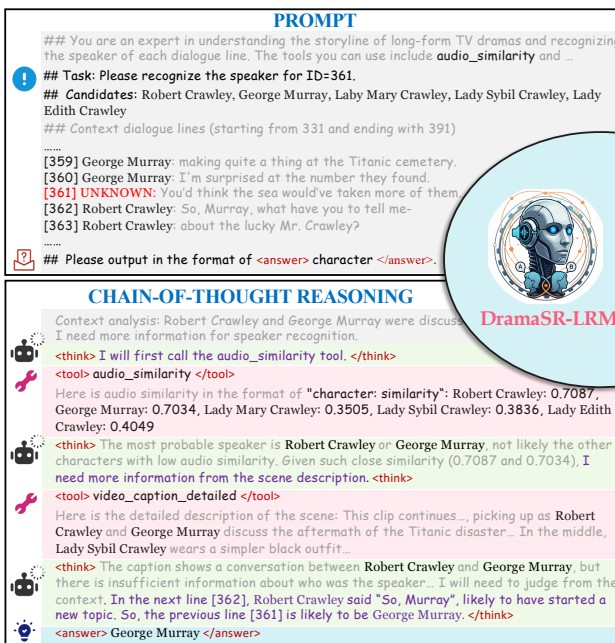

*Figure 2.* An example of chain-of-thought (CoT) reasoning. DramaSR-LRM learns to call different tools (see the ⟨tool_name⟩ and ⟨/tool_name⟩ decorators) and gets feedback from the system (see the texts after ##).

larity between the $n$-th utterance and the $p$-th character's voiceprint set. This value is computed using the mean top-$L$ cosine similarity metric established in our label propagation framework (see Section 4.2).

**Hierarchical video captioning**. We employ a hierarchical captioning strategy. The video is first partitioned into shots using the PySceneDetect library and subsequently aggregated into clips of 10–15 seconds. Given the scale of our data (*e.g.*, a 40-hour drama yields over $10^4$ clips), we process these units in two stages. (a) Base-level captioning: we extract keyframes from each clip and utilize the Qwen3-VL-32B model (Bai et al., 2025a) to generate dense visual descriptions. (b) High-level summarization: We group base clips into semantically cohesive segments (averaging 10 clips per segment) and then utilize the Qwen3-32B model (Yang et al., 2025) to synthesize the clip-level descriptions into a high-level segment summary. Each utterance is thus associated with a dual-context representation: a (detailed) local clip caption and a (brief) global segment summary. Further details are provided in Appendix A.4.

**Character relationship.** To leverage social context and appellations found in dialogue, we maintain a dynamic relational ontology $\mathcal{X}$. By processing the dialogue transcript alongside current speaker attributions, we utilize Qwen3-32B to extract a set of character triplets $(p_1, p_2, \text{relation})$. Here, relation is an open-domain descriptor (*e.g.*, *father*, *colleague*, *friend*) defining the relationship of character $p_2$

to $p_1$. Further details are provided in Appendix A.4.

### 4.4. Training the Large Reasoning Model

The primary objective of the large reasoning model (LRM) is to iteratively refine the initial speaker attributions generated by the label propagation stage. The model is provided with a contextual window of 20–30 dialogue lines and is tasked with synthesizing multimodal evidence to produce a final, high-fidelity attribution. A typical chain-of-thought (CoT) trajectory is illustrated in Figure 2. We adopt a two-stage training paradigm consisting of supervised fine-tuning (SFT) followed by reinforcement learning (RL).

**Data curation.** To establish the model's fundamental tool-use and reasoning capabilities, we curate an SFT dataset using Gemini-3-Pro (Team et al., 2023) as a teacher model. For a given training drama, the teacher model is supplied with the dialogue transcripts and the outputs of the multi-modal toolset (voice_sim, video_cap [both brief and detailed], and char_relation). Gemini-3-Pro is prompted to generate rationales that justify correct attributions by cross-referencing these cues (*e.g.*, 'The voiceprint similarity for Alice is high (0.75)', and 'the caption identifies a female speaker in a lab coat, consistent with Alice's occupation'). To ensure the veracity of the CoT trajectories, we employ a feedback-driven distillation process: if the teacher model produces an incorrect prediction, it is re-prompted with the ground-truth label and instructed to synthesize a refined rationale that logically aligns with the correct answer. Both prompts are detailed in Appendix A.5.

Since the majority of utterances are easy cases (*i.e.*, one character possesses a dominant acoustic similarity), simple classification would not necessitate complex reasoning. To prevent the training set from being overwhelmed by trivial samples, we select a strategic subset (approximately 20% of total utterances) that involves (a) failure recovery: all utterances incorrectly attributed by the label propagation baseline, (b) ambiguity resolution: borderline samples where the margin between the top-1 and top-2 acoustic similarity scores is less than $0.03$[5], and (c) balanced control: a randomly sampled set of clearly correct utterances to balance the dataset.

**SFT training.** We fine-tune a Qwen3-8B model (Yang et al., 2025) on all the curated trajectories, training it to autonomously invoke tools and generate structured reasoning. Another set of samples (with ground-truth labels, but not necessarily with curated trajectories) are reserved for the reinforcement learning phase.

---

[5]Furthermore, to enhance the model's error-correction robustness, we intentionally corrupt 50% of the borderline samples by decreasing the top-1 similarity value by $0.015$ meanwhile increasing top-2 by $0.015$, thereby forcing the LRM to override an initially incorrect acoustic signal using visual or relational reasoning.

| Method | All | En | Cn | Long | Medium | Short | Very Short |
|---|---|---|---|---|---|---|---|
| Facial-aware Guess[†] | 22.54% | 23.30% | 21.75% | 23.21% | 22.43% | 21.53% | 20.78% |
| Label-aware *pyannote* (Bredin, 2023)[‡] | 79.82% | 81.44% | 78.22% | 82.61% | 80.42% | 72.82% | 62.48% |
| Label Propagation (baseline) | 85.49% | 82.41% | 88.58% | 85.34% | 87.12% | 82.37% | 67.45% |
| Qwen3-8B (Yang et al., 2025) (direct use) | 27.40% | 31.66% | 23.17% | 27.45% | 27.26% | 27.62% | 28.65% |
| Qwen3-8B + SFT | 75.22% | 76.21% | 84.60% | 72.63% | 76.67% | 76.97% | 68.61% |
| Qwen3-8B + SFT, w/ conf. sampling | 82.70% | 65.88% | 89.21% | 81.54% | 83.98% | 82.19% | 71.14% |
| DramaSR-LRM (Qwen3-8B + SFT + RL) | 86.93% | 84.94% | 88.91% | 87.45% | 87.77% | 84.12% | **76.95%** |
| **DramaSR-LRM, w/ conf. sampling** | **87.79%** | **85.22%** | **90.37%** | **87.62%** | **88.92%** | **85.70%** | 76.65% |

*Table 2.* Speaker recognition accuracy on the test set (428K utterances). We report accuracy on subsets defined by language **En/Cn** (English/Chinese subsets) and utterance length (**Long**: >2s, **Medium**: 1s–2s, **Short**: 0.5s–1s, **Very Short**: 0s–0.5s. [†]We apply the neighborhood assumption (see Section 4.2) to confirm the candidate speakers. [‡]To introduce character IDs to *pyannote*, we use the Hungarian algorithm to match each recognized speaker with the most-overlapped characters (using the labels) in every 100 dialogue lines.

**RL post-training.** To optimize decision-making under multimodal uncertainty, we employ Group Relative Policy Optimization (GRPO) (Shao et al., 2024). GRPO enhances training efficiency by normalizing rewards across a sampled group of outputs, bypassing the need for a separate value model. Our reward function $R$ is defined by two primary signals, (a) accuracy reward ($r_{acc}$): a binary reward of $+1$ for matching the ground-truth character ID $p_n$, and 0 otherwise, and (b) format reward ($r_{fmt}$): a penalty-based reward to ensure strict adherence to the structured template.

**Iterative inference.** At test time, the LRM operates on the initial labels established during propagation. We employ an iterative refinement loop: if the LRM modifies the attribution for a significant portion of the window, the updated labels are fed back into the context for a subsequent pass. Within this loop, the dynamic tools (`voice_sim` and `char_relation`) are re-computed based on the updated pseudo-labels, while the static, computationally intensive `video_cap` data remains constant. This allows the model to progressively resolve complex multi-character dependencies across the drama.

## 5. Experiments

### 5.1. Experimental Setup

**Datasets and setting.** We partition our benchmark to facilitate a rigorous cross-drama evaluation. SFT is performed on approximately 10K CoT trajectories from *A Lifelong Journey*, while RL utilizes 10K labeled utterances from *Empresses in the Palace*. The model's generalization capabilities are evaluated on the remaining 11 held-out dramas, totaling 428K utterances. These test dramas represent a diverse spectrum of genres (see Table 4 in Appendix A.1), production styles, and acoustic environments. To initialize the label propagation baseline, we provide a sparse seed set containing ground-truth labels for approximately 1% of the utterances for each known character (with a minimum of 1 utterance per character).

**Implementation details.** Our DramaSR-LRM is built upon the **Qwen3-8B** backbone (Yang et al., 2025). The model undergoes 3 epochs of SFT followed by 2 epochs of RL. For the RL phase, we employ GRPO (Shao et al., 2024) with a group size of $G = 8$ and a KL-divergence penalty coefficient of $\beta = 0.0001$. The training was conducted on an 8-node NVIDIA H800 GPU server, requiring approximately 40 hours for completion.

**Confidence sampling.** While our training focuses on 'hard' cases, where top-1 and top-2 acoustic similarities are close, the majority of the test set consists of 'easy' samples. To mitigate the risk of LLM hallucinations on these easy cases, we only invoke the LRM for utterances where the margin between the top-1 and top-2 acoustic scores is less than a threshold $\rho$. This selective reasoning not only improves overall accuracy, but also significantly reduces the computational burden by bypassing LLM on about 80% of test data (for $\rho = 0.10$).

**Computational overhead.** Inference was benchmarked on a single NVIDIA H800 GPU. Acoustic feature extraction is highly efficient, requiring less than 0.01s per utterance. The label propagation baseline, executed on a CPU, takes 2–3 minutes for a drama of 50K utterances. For the LRM, we utilize the vLLM framework to support 256 concurrent threads. This setup achieves an amortized inference time of about 0.33 GPU-seconds per utterance; in other words, an 8-GPU server can finish 50K subtitles within 40 minutes.

### 5.2. Main Results

Table 2 summarizes the speaker recognition accuracy across the benchmark test suite. DramaSR-LRM achieves a substantial performance breakthrough over the competitive label propagation baseline, increasing the mean accuracy from 85.49% to 87.79% (an absolute gain of +2.30%, representing a 16% reduction in the relative error rate). We trained five DramaSR-LRM models individually, and the standard deviation of mean accuracy is 0.39%, much smaller than the

2.30% gain. These results underscore that while acoustic features are foundational, the LRM's capacity to synthesize visual and relational evidence is crucial for resolving identity ambiguity in complex narrative contexts.

**Disclaimer: the impact of the evaluation protocol.** As said in Section 3.3, we applied special treatment for the multi-speaker lines and [UNKNOWN] tags. While this protocol is not final, it does not impact the advantage of DramaSR-LRM over the LP baseline. Specifically: (a) The [UNKNOWN] tag appears rarely, occupying <100 out of 532K (<0.02%) utterances. Excluding these from evaluation reduces both label propagation (LP) and LRM accuracy by only 0.002%. (b) Multi-speaker lines occupy 2K out of 532K (<0.4%) utterances. Since current methods (LP and LRM) treat each line as an inseparable unit, requiring an exact match causes all methods to fail, reducing overall accuracy by 0.34% and 0.35%, respectively. In any choice of the protocol, DramaSR-LRM outperforms LP by 2.3%. We agree that distinguishing multiple speakers is important. In the future, we will train the LRM to detect multi-speaker lines and use an audio separation tool for voiceprint extraction. We can easily build training data by merging adjacent single-speaker lines.

**Comparison to pyannote.** Furthermore, we benchmark our approach against pyannote (Bredin, 2023), a state-of-the-art speaker diarization framework. We map diarization clusters to character IDs using the Hungarian matching algorithm; however, the results indicate that standard diarization algorithms struggle with the long-range temporal dependencies and high character density characteristic of TV dramas. This confirms that the proposed pipeline is perfectly suited for this domain.

**Short utterances.** The right-hand columns of Table 2 highlight a critical failure mode of acoustic-only methods: baseline performance degrades sharply as utterance duration drops below 1.0 or 0.5 seconds. In these 'acoustic sparsity' scenarios, voiceprints are often unstable or non-discriminative. In contrast, DramaSR-LRM maintains robust accuracy, partly via invoking the video_cap and char_relation tools and effectively complements the weakened acoustic signal (see 'tool ablation' in Section 5.3).

**Environmental complexity.** We group the utterances by the number of candidates (*i.e.*, the number of characters in the $\tau = 30s$ neighborhood). In the easy (1–2 candidates, occupying 40.6% of utterances), medium (3–4, 40.8%), and hard (5+, 18.6%) subsets, DramaSR-LRM improves the recognition accuracy of the LP baseline from 84.91%, 86.12%, 85.46% to 87.98%, 87.94%, 86.07%, respectively, showing consistent gains.

**Visual absence.** There are about 9.6K (1.8%) off-screen utterances (*i.e.*, GT speaker's face not appearing in the $\tau =$

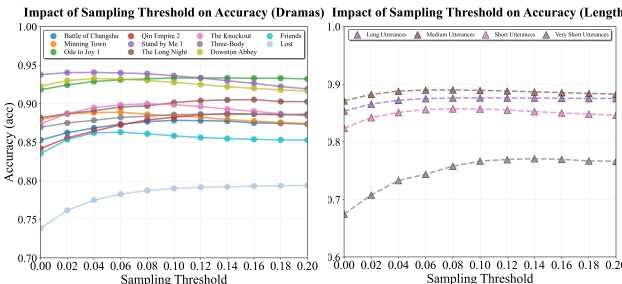

*Figure 3.* Impact of confidence sampling in various TV dramas and the subsets defined by the length of utterances. Full numerical results are provided in Table 5. *Please zoom in for a better view.*

$30s$ neighborhood). To correctly attribute these utterances, the model must assign an [OTHERS] label and then extend the candidate list with 3–5 characters sharing similar voiceprints and reasons again. DramaSR-LRM succeeds in 52.4% of such cases, much higher than the LP baseline that succeeds in merely 13.4% of cases.

**Cross-genre and cross-lingual generalization.** Although the SFT and RL phases utilize data from only two slow-paced Chinese dramas, DramaSR-LRM demonstrates remarkable zero-shot transferability. Significant gains are observed in fast-paced war dramas (*e.g.*, *Qin Empire 2*, +4.06%) and complex crime procedurals (*e.g.*, *The Knockout*, +2.50%), where character interactions are significantly more volatile. Most notably, the model transfers effectively to English-language dramas, delivering significant accuracy improvements (*e.g.*, +2.30% on *Friends* and +5.14% on *Lost*) without specific linguistic adaptation. This suggests that the underlying reasoning patterns for character attribution are language-agnostic.

### 5.3. Diagnostic Experiments

**Tool ablation.** To study the importance of each tool, we ablate the use of tools. We inherit the model trained with the complete tool set, but in the inference stage, when a specific tool is called, the system returns a message saying 'no information can be given'. Results are shown in Table 3. We can draw the following conclusions from the results: (i) voice_sim is the most useful tool, without which the accuracy drops a lot (below the label propagation baseline, which completely relies on this tool), (ii) video_cap also helps, in particular when captions include speaking actions, and (iii) char_relation brings a small gain, but it offers useful clues for hard cases, mostly very short relation-calls (e.g., "Dad" or "Mom"). Additionally, although video_cap and char_relation contribute less to the overall accuracy, they become more important in shorter utterances, where audio features are less stable.

**Impact of confidence sampling.** Figure 3 illustrates the relationship between the sampling threshold $\rho$ and recognition

| Method | All | L | M | S | VS |
|---|---|---|---|---|---|
| Label Prop | 85.49% | 85.34% | 87.12% | 82.37% | 67.45% |
| $-$voice_sim | 72.61% | 72.48% | 73.21% | 70.32% | 62.91% |
| $-$video_cap | 86.76% | 86.45% | 87.81% | 83.30% | 72.33% |
| $-$char_rela | 87.55% | 87.41% | 88.68% | 85.33% | 75.66% |
| Full Set | 87.79% | 87.62% | 88.92% | 85.70% | 76.65% |

*Table 3.* Speaker recognition accuracy on the test set, with each tool ablated (two caption tools are combined) during the inference stage. Mind the impact of each tool *w.r.t.* utterance length.

accuracy (full numerical results in Tables 5 and 6). Our results indicate that the optimal $\rho$ is inversely correlated with the fidelity of the initial label propagation. For dramas with high initial accuracy (*e.g.*, *Stand by Me*, baseline 93.77%), a conservative threshold ($\rho = 0.04$) is preferable, invoking the LRM only for highly ambiguous cases. Conversely, for dramas where acoustic cues are less reliable (*e.g.*, *Lost*, baseline 73.89%), a larger threshold ($\rho = 0.20$) allows the LRM to intervene more frequently, correcting systematic errors in the propagation. This trend is also observed in subsets with 'acoustic sparsity' (short and very short utterances), which consistently demand higher $\rho$ values to achieve performance gains. *To provide a generalized solution across the entire benchmark, we fix $\rho = 0.1$ for all main experiments.*[6]

**Iterative reasoning.** Since DramaSR-LRM utilizes context speaker labels as part of its reasoning input, it is inherently compatible with an iterative refinement loop as detailed in Algorithm 1. Evaluate on the drama *The Long Night*, the first pass improves the label propagation baseline from 88.18% to 90.15% (+1.97%), while a second pass further elevates performance to 90.40% (+0.25%). This demonstrates that the LRM can 'self-evolve' by re-evaluating its own previous outputs. *However, to maintain a favorable trade-off between accuracy and computational latency, we report results using only a single iteration in the main experiments.*

**Robustness to contextual variation.** DramaSR-LRM exhibits robustness to the length of the dialogue context window. We tested the model (trained on 30-line contexts) across varying window sizes, and found that the overall accuracy fluctuations do not exceed 0.2%. Furthermore, since the dialogue text represents a relatively small fraction of the total token count compared to the dense multimodal tool logs and long CoT trajectories, adjusting the window size does not impose a prohibitive computational overhead on inference time. *Consequently, we employ a standardized 30-line context for all main experiments.*

---

[6]We also test different $\rho$ values in the 10% validation set of the RL data, and $\rho = 0.1$ reports the best validation accuracy.

### 5.4. Downstream Utility: Video Understanding

To evaluate the cascading benefits of accurate speaker recognition on holistic video understanding, we perform downstream assessments on video captioning and video question-answering. We provide a state-of-the-art MLLM (Qwen3-VL-32B (Bai et al., 2025a)) with video clips paired with two transcription variants, produced label propagation and DramaSR-LRM. An example is illustrated in Figure 12, where the integration of DramaSR-LRM's labels improves the factual grounding of generated descriptions. This is particularly evident in resolving complex narrative queries, such as 'who claimed what', where the model must distinguish between subtle character motivations and conflicting dialogue. These results confirm that speaker attribution is not merely an auxiliary task, but a foundational prerequisite for deep, character-centric narrative comprehension.

To quantitatively study the question-answering (QA) accuracy, we collected 20K+ QA pairs (for plot understanding) on these 13 dramas. In the subset of 18,399 QAs (covering 11 dramas in the SR test set), we used Qwen3-VL-32B for multimodal QA, and different speaker recognition results (*i.e.*, no labels, label propagation, DramaSR-LRM, and ground-truth) are used as auxiliary clues. The QA accuracy under these four settings are 21.6%, 70.3%, 72.0%, and 80.8%, respectively. These results not only show the importance of speaker recognition in video QA, but also demonstrate the downstream gain of our method, as well as the room for improvement (compared to GT).

## 6. Conclusion

In this paper, we have addressed the formidable challenge of speaker recognition within the context of long-form TV dramas. We introduced **DramaSR-532K**, a comprehensive benchmark comprising 532K annotated utterances across 13 distinct dramas, providing the research community with a high-fidelity resource. We proposed **DramaSR-LRM**, a large reasoning model that moves beyond isolated acoustic biometrics by autonomously synthesizing multimodal evidence through autonomous tool-use. Our empirical evaluations demonstrate that DramaSR-LRM significantly outperforms the label propagation baseline, particularly in the scenarios such as acoustic sparsity, secondary characters, high character density, and off-screen dialogue. Furthermore, we validated that improved speaker recognition directly translates to superior performance in downstream video understanding tasks. By releasing DramaSR-532K and our reasoning framework, we provide a foundation for future research into character-centric video analysis. In the future, we will investigate more challenging settings, including **open-world** speaker description and **end-to-end** speaker recognition, *i.e.*, operating directly on raw audio signals rather than pre-segmented dialogue transcriptions.

## Impact Statement

This work introduces DramaSR-532K, a benchmark for speaker recognition in long-form TV dramas, and DramaSR-LRM, a multimodal reasoning framework. While the primary focus of this research is to advance the state-of-the-art in narrative video understanding, we recognize several broader implications:

- **Large reasoning models (LRMs).** As this paper explores the use of LRMs for evidence synthesis, it contributes to the ongoing discussion regarding the reliability and transparency of AI reasoning. Our use of verifiable chain-of-thought trajectories and tool-use logs is a step toward making multimodal AI decisions more interpretable.

- **Content accessibility.** Precise speaker recognition is a foundational technology for generating high-quality automated closed-captioning and audio descriptions. By improving attribution in complex, multi-character scenarios, this work can contribute to more accessible media experiences for the hearing-impaired and visually-impaired communities.

- **Ethical use of data.** The DramaSR-532K benchmark is derived from publicly available television dramas. We have taken care to ensure that the data is used for non-commercial research purposes. However, we acknowledge that advancements in facial recognition and voiceprint analysis must be handled with caution regarding privacy. We advocate for the use of these technologies in accordance with established ethical guidelines and legal frameworks.

As a standard research contribution in video content analysis, this work does not present any immediate societal risks beyond those typically associated with the development of large-scale language, audio, and vision models.

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

| Lang. | Series' Name (Local Name) | Genre | #S. | #Ep. | Duration | #char. | #uttr. |
|---|---|---|---|---|---|---|---|
| En | *Downton Abbey* | Drama | 6 | 52 | 51h,08mins | 144 / 435 | 56,583 |
| | *Friends* | Comedy | 10 | 236 | 89h,25mins | 70 / 744 | 95,564 |
| | *Lost* | Triller | 5 | 121 | 87h,33mins | 77 / 709 | 66,688 |
| | **Subtotal** | – | 21 | 409 | 228h,06mins | 291 / 1888 | 218,835 |
| Cn | *A Lifelong Journey* (人世间) | Drama | 1 | 58 | 44h,09mins | 91 / 881 | 50,729 |
| | *Battle of Changsha* (战长沙) | War | 1 | 32 | 23h,31mins | 34 / 612 | 20,680 |
| | *Empresses in the Palace* (甄嬛传) | Palace | 1 | 76 | 57h,10mins | 99 / 565 | 54,060 |
| | *Minning Town* (山海情) | Rural | 1 | 23 | 17h,34mins | 85 / 509 | 23,932 |
| | *Ode to Joy 1* (欢乐颂1) | Romance | 1 | 42 | 30h,04mins | 59 / 456 | 35,949 |
| | *Qin Empire 2* (大秦帝国之纵横) | Historical | 1 | 51 | 38h,42mins | 58 / 744 | 29,127 |
| | *Stand by Me 1* (一起同过窗1) | Teen | 1 | 34 | 25h,18mins | 30 / 355 | 33,749 |
| | *The Long Night* (沉默的真相) | Suspense | 1 | 12 | 10h,20mins | 37 / 216 | 8,263 |
| | *The Knockout* (狂飙) | Crime | 1 | 39 | 27h,02mins | 74 / 883 | 36,183 |
| | *Three-Body* (三体) | Sci-Fi | 1 | 30 | 23h,35mins | 45 / 403 | 20,949 |
| | **Subtotal** | – | 10 | 397 | 297h,25mins | 612 / 5624 | 313,621 |
| **Total** | – | – | 31 | 806 | 525h,31mins | 903 / 7512 | 532,456 |

*Table 4.* A list of all long-form TV dramas used in this study. The dramas in each sub-category (English/Chinese) are sorted in alphabetical order by their English name. We count the number of known/all characters for each drama.

## A. Details

### A.1. Data Preparation

**Video Pre-processing.** TV dramas used in this paper are displayed in Table 4. Each drama in our dataset comprises multiple episodes. To ensure narrative continuity, we first excise the opening and closing sequences (titles and credits) of each episode. The remaining footage is concatenated into a single, continuous video stream for each drama. This concatenation provides extended temporal context for dialogue occurring at episode boundaries, which significantly assists the DramaSR-LRM in maintaining character consistency and inferring speakers through broader narrative flow.

**Transcription extraction.** We extract dialogue text from hard-rendered subtitles using an OCR and refinement pipeline.

- **Initial subtitle extraction.** For each drama, we define a spatial bounding box encompassing the subtitle region to minimize background noise. We then utilize PaddleOCR-v4 (Cui et al., 2025b;a) for frame-by-frame recognition, yielding a raw text sequence $\{\mathbf{t}_n\}$.

- **Grouping texts into utterances.** To aggregate consecutive frames into discrete dialogue lines, we compute the normalized Levenshtein similarity (*a.k.a.*, 1 minus edit distance) between adjacent frames:

$$\text{sim}(\mathbf{t}_n, \mathbf{t}_{n+1}) = 1 - \frac{\text{Levenshtein}(\mathbf{t}_n, \mathbf{t}_{n+1})}{\max(|\mathbf{t}_n|, |\mathbf{t}_{n+1}|)}. \tag{1}$$

  Frames with a similarity score exceeding $0.8$ are grouped into a single utterance.

- **MLLM refinement.** For each identified segment, the median frame is processed by Qwen3-VL-32B (Bai et al., 2025a). This step corrects OCR artifacts, resolves text segmentation for overlapping subtitles, and enforces linguistic formatting consistency.

- **Quality control.** We perform manual verification on approximately $1\%$ of the corpus, specifically targeting anomalies where the text length is disproportionate to the temporal duration or where contextual logic appears fractured.

**Character library construction.** To achieve high-fidelity facial reference data, we implement a two-stage construction and enrichment pipeline that integrates external and *in-situ* visual information:

- **Initial face library.** Cast metadata and reference portraits are harvested from public databases (*e.g.*, IMDB and TMDB for English dramas, Douban and BaiduBaike for Chinese dramas). We perform manual cross-verification to establish a definitive mapping between actors and their respective in-drama roles.

- *In-situ* **library enrichment.** To account for variations in lighting, aging, and makeup within the drama, we scan the entire video at 1 FPS. Using the initial library as a reference, we extract all matching facial instances. To manage the scale and quality of this data, we apply a sequential filtering pipeline:

  - **Multi-face exclusion.** Samples containing multiple faces are discarded to avoid identity ambiguity.
  - **Deduplication.** We remove redundant samples using a cosine similarity threshold of $0.8$.
  - **Quality filtering.** Low-resolution or heavily occluded faces are removed based on a quality threshold of $0.45$.
  - **Quality control.** From the remaining high-quality pool, we randomly sample $15$ representative images per character to form the final reference library.

- **Frame-wise scan.** Finally, we utilize the InsightFace library (Deng et al., 2019) to perform frame-by-frame detection and recognition across the full duration of each drama, establishing the foundational visual evidence for speaker attribution.

### A.2. Speaker Label Annotation

Human annotators are provided with three primary information streams via a specialized graphical user interface (GUI): (1) video streams with overlaid facial detection bounding boxes, (2) initial speaker hypotheses generated by the label propagation algorithm, and (3) acoustic biometric statistics (*e.g.*, voiceprint similarity scores). Annotators are instructed to watch the videos sequentially and perform corrective edits on the initial labels as necessary.

---

**Guideline for human labelers: annotation taxonomy and hierarchy.** For each dialogue segment, the annotator must first categorize the utterance based on the number of active speakers.

1. **Single-speaker utterances.**
   (a) **Primary characters.** If the speaker exists in the established character library, the annotator must use the unique canonical name.
   (b) **Ancillary or open-set characters.** If the speaker is identifiable but not in the library, the following naming priority is enforced:
      i. **Nomenclature.** Use full or partial names if revealed in the dialogue (*e.g.*, *Jack* or *Mr. Smith*).
      ii. **Professional/social role.** Use inferred occupations or social status (*e.g.*, *Doctor*, *Director*).
      iii. **Acousmatic sources.** For off-screen voices, provide a descriptive source (*e.g.*, *Reporter*, *Digital Voice*). Specific narrative markers like *Narration* or *Interlude* are used where appropriate.
      iv. **Relational identifiers.** Use kinship or hierarchy relative to primary characters (*e.g.*, *Alice's Daughter*).
      v. **Visual descriptors.** If no other information is available, use salient physical traits (*e.g.*, *Man in Black*, *Short-haired Woman*).
      vi. *Numerical indexing.* Distinguish multiple similar unknown characters using indices (e.g., *Worker 1*, *Worker 2*, *etc.*).
   (c) **Unidentifiable.** Use the [UNKNOWN] tag if the identity remains ambiguous after multimodal review.

2. **Simultaneous multi-speaker utterances.**
   (a) **Group attribution.** If $P' \leqslant 5$ speakers are identifiable, provide a set of all individual names.
   (b) **Choral/mass speech.** If $P' > 5$ speakers are present or the speech is choral, assign the [MULTIPLE] tag. This can be combined with individual names if specific primary characters are audible within the crowd (*e.g.*, *Alice* and *Bob* and [Multiple] *Students*).

3. **Sequential composite utterances.** For rapid-fire interruptions or multi-speaker segments within a single line, the annotator must split the segment and attribute each sub-unit independently.

---

*Figure 4.* A brief version of the annotation guideline for labelers.

To ensure high-fidelity labeling, particularly for ancillary roles that lack pre-existing voiceprints, we implement a stringent quality control protocol. Following the initial annotation, an automated script flags high-entropy labels for second-pass

human verification. A label is marked for review if it meets any of the following criteria: (a) a primary character assigned to an utterance with low acoustic similarity to their reference set, (b) a primary character speaking while off-screen, (c) an newly introduced ancillary character, or (d) an [UNKNOWN] attribution.

On average, the primary annotation phase requires 1.5–2.5 hours per video hour, while the targeted quality-control validation requires an additional 10–15 minutes per video hour. This tiered approach minimizes errors in the ground truth, particularly for the difficult edge cases analyzed in Section 5.

### A.3. Label Propagation

**Acoustic feature extraction.** We utilize the ffmpeg library to extract raw audio from the video containers at a sampling rate of 16kHz. For each dialogue segment $n$ in the set $\mathcal{S} = \{(\mathbf{t}_n, a_n, b_n)\}_{n=1}^N$, we isolate the corresponding audio clip using the pydub library. Voiceprint embeddings are generated via the ERes2Net model (Chen et al., 2023) within the **3D-Speaker** toolkit (Chen et al., 2025). To ensure a stable similarity metric, all extracted feature vectors are $\ell_2$-normalized, mapping them to a hyperspherical space where cosine similarity is equivalent to the dot product.

**Seed cluster generation.** Given the candidate utterance sets $\{\mathcal{S}_p\}$ associated with character $p$ via the neighborhood assumption, we initialize a high-confidence clustering procedure controlled by a similarity threshold $\theta_{\text{seed}}$ (initially 0.85, a high value to guarantee that two utterances with a higher similarity belong to the same character) and a minimum cluster size $\eta_{\text{seed}}$. Then, we employ a two-phase greedy search strategy:

- **Phase 1: Global high-confidence search.** For all active characters, we construct an acoustic graph where an edge exists between two nodes (utterances) if their similarity is greater than $\theta_{\text{seed}}$. We identify the maximum connected component (MCC) for each character using a floodfill algorithm. The MCCs are ranked by cardinality. If the largest MCC meets the $\eta_{\text{seed}}$ requirement and maintains sufficient separation from existing seeds (similarity difference is greater than $\Delta\theta_{\text{seed}}$), it is accepted. If no MCC is accepted, we iteratively decay $\theta_{\text{seed}}$ (step size 0.01) until a lower bound of 0.70 is reached.

- **Phase 2: Targeted character search.** For any character $p$ lacking a seed after Phase 1, we perform a local MCC search within $\mathcal{S}_p$. We apply the same validation criteria against existing clusters to prevent identity collision. The threshold $\theta_{\text{seed}}$ is gradually relaxed until every active character is associated with one seed voiceprint set.

This process includes an optional human-in-the-loop verification to ensure the purity of the seed sets. For a typical drama involving approximately 50 characters, this supervised seed generation requires roughly 2–3 hours.

**Iterative affinity propagation.** Once seeds are established, we expand the attributions to the entire corpus through an iterative propagation strategy. Starting from a high threshold $\theta_{\text{prop}}$, we repeat a dual-pass search:

- **Identity expansion.** Each unlabeled utterance is compared against all existing character clusters. If the maximum similarity exceeds $\theta_{\text{prop}}$, the utterance is assigned to that character and integrated into the reference set, dynamically updating the cluster centroid.

- **Latent identity discovery.** Within a sliding window of 20 dialogue lines, we search for MCCs using a more stringent threshold ($\theta_{\text{prop}} + \Delta\theta_{\text{prop}}$). MCCs exceeding $\eta_{\text{prop}}$ in size are initialized as new ancillary-character clusters[7]. This allows the model to track consistent voices that were not captured by the initial visual-anchor sets.

We decrement $\theta_{\text{prop}}$ by 0.02 per iteration. The process terminates at $\theta_{\text{prop}} = 0.45$. Any utterances remaining unassigned at this stage are designated with the [UNKNOWN] tag, pending final adjudication by the LRM.

### A.4. Multimodal Toolset

**Clip segmentation via adaptive merging.** We utilize the PySceneDetect library to perform initial content-aware shot segmentation. Using an adaptive thresholding algorithm that monitors visual intensity and color histogram shifts, we partition the video into discrete shots with an average duration of 2–3 seconds.

---

[7]While these ancillary clusters are treated as unique identities capable of propagating labels to other utterances, they lack grounding in the visual reference library. Consequently, to prevent identity drift, the propagation of these labels is restricted to a local temporal neighborhood of the original cluster, unlike primary characters who benefit from global facial anchors.

---

**The video clip description prompts**

Task You are a video understanding expert. Please create a description for the current video clip (containing multiple frames arranged in chronological order).

Clip Description Guidelines

1. **Character Description Guidelines**:

   (a) When referring to a character, use the name displayed above the red box around the character's face in the video frames. However, do not directly mention terms like 'red box' in the description.

   (b) If a character has no bounding box or name above the face, distinguish and describe them based on their appearance features, such as clothing style or other prominent characteristics.

   (c) When a character appears for the first time, describe their features, such as clothing and hairstyle.

2. **On-screen Text Description Guidelines**:

   (a) If text appears in the video frames, describe the text in its original language. If the recognized result is subtitles, it needs to be aligned with the additionally provided dialogue information.

   (b) If there are objects with large sections of text in the frame, such as letters or notices, add the symbol ⟨texts⟩ in the output description. For example: 'He is holding a letter ⟨texts⟩, which says...'.

3. **Other Detail Description Guidelines**:

   (a) **Note**: Please provide as many details as possible, including the characters' movements, behaviors and expressions, interactions between characters, as well as details like the color and texture of the objects being manipulated.

   (b) **Note**: The description can include the location where the story takes place, environmental details, weather characteristics, and important objects that reflect the background of the era.

Reference Example Assume the input information includes multiple video frames.

Output Description: {'Clip Level Description': 'The clip depicts an outdoor scene on a cold winter day, on a dilapidated street with thick snow hanging from the eaves. The young boy Guangming is wearing XXX clothes and an XXX hat, with his cheeks flushed from the cold and snot dripping down. Zhou Bingkun is dressed in blue XXX clothes and an XXX hat, with a serious expression. The young boy Guangming pulls him, and Zhou Bingkun turns around and stops.'}

Output Format: Your response should be in the following format: {'Clip Level Description': 'The clip XXX'}

---

*Figure 5.* The prompt to create detailed chronological descriptions for video clips based on multi-frame visual information with specific character and detail description guidelines.

To aggregate these shots into semantically cohesive clips, we employ a multi-view representation strategy. For each shot, we extract the first, middle, and last frames and compute their embeddings using the CLIP ViT-L model (Radford et al., 2021). The concatenation of these three vectors forms a robust representation of the shot's visual content. We then calculate pairwise cosine similarities between adjacent shots. To account for stylistic variations across different dramas, we apply a dynamic quantile thresholding strategy: the merging threshold is defined as the top $1/5$ quantile of all similarity scores within a specific drama. Adjacent shots exceeding this threshold are merged until a target duration of 8–15 seconds is reached. This process ensures that clips are long enough to provide narrative context but short enough to be processed efficiently by the multimodal model.

**Base clip captioning.** Let the generated clip sequence be $V = (C_1, \ldots, C_L)$. For each clip $C_l$, we sample frames at 1 FPS, prioritizing frames with high facial density. We represent each clip as a sequence of these sampled frames and utilize Qwen3-VL-32B (Bai et al., 2025a) for caption generation. To enhance the grounding of these captions, we implement two visual-linguistic augmentations: (a) Textual anchoring: We integrate the dialogue transcript and current speaker pseudo-labels directly into the model's context, and (b) Visual overlay: we overlay red bounding boxes on detected faces, labeled with the corresponding character ID. This results in detailed descriptions averaging 300 words per clip, ensuring that the visual narrative is tightly coupled with character identities. Figure 5 shows the prompts for video clip captioning.

**Hierarchical Segment Summarization.** To manage the computational overhead of processing millions of words per

drama, we aggregate clips into consecutive sections $S_m$ via semantic partitioning. We extract text embeddings for each clip caption using the BAAI's bge-large-zh-v1.5 model (Xiao et al., 2024), and compute the cosine similarity between temporal neighbors $(C_l, C_{l+1})$. Clips are merged into a section if their caption similarity exceeds $0.8$, subject to a constraint on the section size ($8 \pm 4$ clips).

For each resulting section $S_{l'}$, Qwen3-32B (Yang et al., 2025) is tasked with synthesizing the constituent clip captions into a single, high-level segment summary $\mathbf{T}_{l'}$. This summary filters redundant visual details and focuses on core plot developments and character interactions. Figure 6 shows the prompts for video segment summarization.

---

**The video summarization prompts (datailed version)**

You are an expert in understanding descriptions of video clips, where the video consists of {number_of_sub} clips. Your task is to summarize these clip descriptions in chronological order to create a coherent video section description.

Detailed Description Guidelines for Video Section Description

1. Since clip descriptions are provided in chronological order, ensure the description is coherent and follows the same sequence. Do not refer to the first/last frame of any individual clip as that of the entire video.

2. The clips are continuous; pay attention to maintaining logical coherence when summarizing.

3. **Note:** Present text and dialogue from the clips in a paraphrased summary form. Avoid frequent use of expressions like 'XXX said: '...'; instead, prefer 'XXX did something and expressed XXX meaning'.

4. **Note:** Due to clip segmentation, errors may exist. Correct such errors when merging clip descriptions, ensuring smooth narration at the junctions—for example, merge statements from the same person or descriptions of the same person/object. If subtitles for the entire section are provided, use their context to make judgments; only correct errors when confident, and do not make unfounded assumptions.

5. **Note:** Merge repeated descriptions of a character's expressions, appearance, or state (e.g., 'appearing helpless and powerless', 'wearing a blue coat with a red badge on the chest', 'continuing to wash dishes on the other side of the room without participating in the conversation') to avoid redundancy. Merge duplicated dialogue only if it is described twice for the same clip (not if the character actually spoke twice).

6. **Note:** The tone of the video description should mimic direct narration of a video, not a summary of information from multiple clips. Thus, avoid expressions from the reference clip descriptions such as 'this clip begins...', 'as the clip progresses...', 'this clip ends', 'the final/initial frame', 'the second clip starts with...', 'the last few frames of this part'.

7. **Note:** Incorporate all details from the given clip descriptions, but avoid repeating descriptions of the same shot. Try to understand the video's theme and provide a coherent narrative that connects all clips.

8. **Note:** If subtitles or character relationships for these clips are provided, use them to aid understanding and correct errors.

9. **Note:** Since the duration of each clip varies, the length of the concise description must meet the specific word count requirement for each case. Retain as much information as possible while reducing detailed descriptions.

Output Format: Your response must follow the following structure: {'Section Detailed Description': 'The section ......'}

---

*Figure 6.* Video summarization prompts for consolidating individual clip descriptions into a unified, detailed narrative of the entire section.

**Character relationship extraction.** To capture the evolving social dynamics inherent in drama series, we implement a temporal relational ontology. We process the transcripts and speaker labels episode-by-episode, prompting Qwen3-32B to extract character triplets $(p_1, p_2, \text{relation})$. These episode-wise relational lists are then consolidated into a global mapping where each relationship is timestamped (*e.g.*, *Alice* is *Bob's colleague* in Episodes 1–10; *Alice* is *Bob's rival* in Episodes 11–20). This temporal awareness allows the LRM to resolve identity ambiguities that arise as characters' social roles shift over the course of the storyline. Figure 8 shows the prompts for character relationship extraction.

---

**The video summarization prompts (From the detailed version to the brief version).**

You are an expert in understanding video clip descriptions. Your task is to summarize a concise description of the video based on its detailed description, and generate a corresponding title.

Guidelines for Concise Description

1. The concise description is a summary of the detailed description.

2. It must include key elements such as people/objects involved, actions performed, locations, and core events. Retain information useful for understanding the plot, but omit excessive detailed descriptions.

3. It should contain distinguishing features of the scene, such as the story's setting, unique plot points, or main character relationships.

4. If subtitles or character relationships of the video are provided, use them to check and correct any errors.

5. **Note:** Retain as much key information as possible while minimizing redundant details.

Guidelines for Title

1. The title should be in the form of a phrase, concisely capturing the core event of the video.

Output Format: Your response must follow the following structure: {'Section Brief Description': 'The section XXXX', 'Title': 'XXX' }

---

*Figure 7.* The prompt to condense the description of the entire video segment into a concise version.

---

**The character-relationship-graph extraction prompts.**

You are an AI assistant specializing in extracting character relationship graphs from text. Your task is to carefully read the text provided by the user, identify the characters mentioned therein, as well as their explicit or strongly implied stable relationships (such as kinship relationships, work relationships, and social relationships, such as father-son, mother-son, elder sister-younger brother, teacher-student, friend, colleague, superior, subordinate, lover, enemy, etc.). Temporary event interactions like 'meeting' or 'helping' should not be included.

Key Notes

1. Ensure the accuracy and consistency of character names throughout the JSON. If the user provides a list of character names, only extract the relationships among the specified characters.

2. The relationships in 'relationships' should be directional. Pay attention to the directionality or superior-subordinate relationship from 'Character 1' to 'Character 2'. For example, for a teacher-student relationship, it should be formatted as ['Teacher's Name', 'Student's Name', 'teacher-student'] or ['Student's Name', 'Teacher's Name', 'student']. For a mother-son relationship, it should be ['Mother's Name', 'Son's Name', 'mother-son'] or ['Son's Name', 'Mother's Name', 'son'], with specific roles or directions reflected in the relationship description.

3. Only extract stable relationships that are explicitly stated in the text or can be reasonably inferred. Ignore temporary interactions or mentions of characters with no clear relationships.

4. The final output must be a strictly compliant JSON object. Ensure the JSON format is correct and error-free, and do not include any explanations, comments, or code markers other than the JSON content.

5. The text may contain dialogues between characters without specifying the speakers. Please infer the speakers based on the context.

Please return the extracted relationship graph in JSON format. The JSON object must contain two keys: 1. 'characters': A list containing all the names of characters involved in the relationships. Ensure there are no duplicates or omissions, and the names remain consistent throughout. 2. 'relationships': A list containing relationship triples. Each triple is a list in the format of ['Character 1', 'Character 2', 'Relationship Description'].

---

*Figure 8.* The prompt to extract character relationship graphs from text and output in standard JSON format.

## A.5. Data Curation

We utilize Gemini-3-Pro (Team et al., 2023) as the teacher model to curate our SFT trajectories. The system prompts governing the model's tool-use behavior are detailed in Figures 9 and 10. During the generation phase, the model is

**The prompts for SFT data curation (Part I).**

You are an intelligent assistant responsible for speaker recognition of lines in film and television drama scenes. The user will first provide a list of candidates, the serial number of the target line, and the context of the target line. Please judge the speaker of the target line in the drama based on the contextual information initially provided by the user and the additional information obtained by calling the available tools introduced below. You need to select the most suitable speaker from the given list of candidates.

Tool Introduction

1. `audio_sim`: No parameters are required to call this tool. The tool will output the audio similarity between the voice library of each real candidate and other possible characters in the scene and the voice of the target line.

2. `video_cap_detailed`: No parameters are required to call this tool. The tool will output a textual description of the drama frames within a few seconds before and after the target line is spoken, through which you can understand the frame details of the moment the target line is delivered.

3. `video_cap_brief`: No parameters are required to call this tool. The tool will output a textual description of the story plot where the target line is located, through which you can understand the short-term story background of the target line's occurrence.

4. `char_relation`: No parameters are required to call this tool. The tool will output all character relationships of each real candidate in the drama. **Key Note**: The character relationships provided by the tool are not limited to those existing when the target line is spoken, but include all relationships that have appeared between characters throughout the entire drama. That is, some of the character relationships provided by the tool may be valid at the time the target line is delivered, while others may be valid in the past or future of that moment. For example, Alice and Bob are classmates and friends now, but will become lovers as the plot develops, and the tool will output these three relationships at the same time.

Guidelines for Tool Calling and Result Output

1. You must always output in Chinese text format and use specific identifiers to separate the content of each part in the output. Specifically, you need to use ⟨think⟩ and ⟨/think⟩ as the start and end identifiers to mark your thinking process, ⟨tool⟩ and ⟨/tool⟩ to mark your tool calls, and ⟨answer⟩ and ⟨/answer⟩ to mark your final speaker recognition result.

2. You must choose either tool calling or final result output for each response. If you need to call tools to obtain more information, use ⟨tool⟩ and /tool⟩ to mark the tool call information and do not output the result with answer⟩ and ⟨/answer⟩ at this time. If you decide to give the final character judgment, use ⟨answer⟩ and ⟨/answer⟩ to mark the result output and do not mark the tool call with ⟨tool⟩ and ⟨/tool⟩ at this time.

3. When marking tool call information with ⟨tool⟩ and ⟨/tool⟩, the marked content is the name of the tool you need to call. If parameters are required for the tool call, you need to place the parameters in English parentheses after the tool name in sequence and separate them with English commas in accordance with the tool instructions (i.e., *ToolName(param1,param2,...)*). If no parameters are required, you only need to output the tool name within the separators. After you call a tool, the user will provide the result returned by the called tool in the next message.

4. When marking the final result information with ⟨think⟩ and ⟨/think⟩, the marked content is the name of the final speaker you give, i.e., the name of the character corresponding to the target line (i.e., `Character Name`). Once you give the final result, the user will receive the result and end the conversation.

5. Do not output any content that is not within any pair of identifier pairs.

*Figure 9.* The prompt to identify the speaker of the target line based on contextual information and multi-tool invocation.

initialized with the user prompt described in Figure 11, which incorporates the initial speaker labels. To ensure high-quality reasoning, we implement a two-pass fallback mechanism: the 'cheat message' is withheld during the initial inference. If the model fails to produce a valid or consistent result in the first pass, the cheat message is provided for a secondary re-evaluation. Empirical observation suggests that this iterative refinement allows the model to successfully converge on the

---

**The prompts for SFT data curation (Part II).**

Emphasized Notes

1. Lines with consecutive IDs are also consecutive in the drama plot, and the provided lines are excerpted from the drama, so the provided lines may not represent a complete dialogue scene. In addition, the provided lines do not exactly constitute the complete story corresponding to `video_cap_brief`, and there may be differences in their context ranges. You can infer the characters present in the scene based on the line context, the frame description in `video_cap_detailed`, and the plot description in `video_cap_brief`. Combined with the two types of description information, you can reason based on the contextual context, character dialogue relationships, character identity relationships, personality, speaking style, etc., and finally judge the speaker of the target line in the drama.

2. If there are address terms in the target line or the contextual lines that form a dialogue relationship with the target line, or the expression of the target line and its context implies the relationships between characters in the scene, you can actively try to call the `char_relation` tool (Tool 4) to obtain relevant relationship information.

3. Please fully invoke the tools before outputting the answer, and try not to give the answer directly in the first response.

4. Only one tool can be invoked per output.

5. Please ensure that your thinking process marked with ⟨think⟩ and ⟨/think⟩ is included in every output. (This requirement is emphasized three times: ensure the thinking process is included in every output; ensure the thinking process is included in every output; ensure the thinking process is included in every output.)

---

*Figure 10.* (Continuing Figure 9) The prompt to identify the speaker of the target line based on contextual information and multi-tool invocation.

correct speaker attribution in the vast majority of cases.

## B. Additional Results

| Lang. | Name | LP | DramaSR-LRM | | | | | | | | | |
|---|---|---|---|---|---|---|---|---|---|---|---|---|
| | | | 0.02 | 0.04 | 0.06 | 0.08 | 0.10 | 0.12 | 0.14 | 0.16 | 0.18 | 0.20 |
| En | *Downton Abbey* | 92.28 | 93.00 | **93.27** | 93.19 | 93.02 | 92.77 | 92.49 | 92.21 | 91.98 | 91.80 | 91.63 |
| | *Friends* | 83.53 | 85.36 | 86.22 | **86.31** | 86.09 | 85.83 | 85.61 | 85.48 | 85.38 | 85.31 | 85.27 |
| | *Lost* | 73.85 | 76.17 | 77.49 | 78.27 | 78.75 | 79.01 | 79.17 | 79.22 | 79.33 | 79.35 | **79.40** |
| | **Subtotal** | 82.41 | 84.15 | 85.02 | 85.29 | **85.31** | 85.22 | 85.10 | 84.99 | 84.93 | 84.87 | 84.83 |
| Cn | *Battle of Changsha* | 85.29 | 86.23 | 86.85 | 87.35 | 87.67 | **87.84** | 87.80 | 87.75 | 87.51 | 87.46 | 87.34 |
| | *Minning Town* | 87.93 | 88.67 | 88.85 | **88.91** | 88.61 | 88.44 | 88.23 | 87.95 | 87.74 | 87.56 | 87.41 |
| | *Ode to Joy 1* | 91.81 | 92.43 | 92.88 | 93.10 | 93.23 | **93.34** | **93.34** | 93.33 | 93.31 | 93.31 | 93.22 |
| | *Qin Empire 2* | 84.18 | 85.56 | 86.44 | 87.26 | 87.89 | 88.24 | 88.58 | **88.71** | 88.66 | 88.56 | 88.64 |
| | *Stand by Me 1* | 93.77 | 94.03 | **94.06** | 94.00 | 93.89 | 93.63 | 93.34 | 92.98 | 92.64 | 92.23 | 91.91 |
| | *The Long Night* | 88.18 | 88.72 | 89.08 | 89.57 | 89.72 | 90.15 | 90.39 | 90.49 | **90.53** | 90.28 | 90.27 |
| | *The Knockout* | 87.37 | 88.73 | 89.49 | 89.85 | **89.99** | 89.87 | 89.63 | 89.30 | 89.02 | 88.70 | 88.43 |
| | *Three-Body* | 86.95 | 87.50 | 87.85 | 88.22 | 88.37 | 88.58 | 88.63 | 88.60 | **88.66** | 88.60 | 88.51 |
| | **Subtotal** | 88.58 | 89.41 | 89.88 | 90.20 | 90.33 | **90.37** | 90.32 | 90.18 | 90.03 | 89.85 | 89.71 |
| | **Total** | 85.49 | 86.78 | 87.45 | 87.74 | **87.82** | 87.79 | 87.71 | 87.59 | 87.48 | 87.36 | 87.27 |

*Table 5.* Drama-wise speaker recognition accuracy (%) with respect to different confidence sampling thresholds. The dramas are sorted in the same order as in Table 4. We have used $\rho = 0.10$ consistently in the main experiments (Table 2). **LP**: label propagation (*i.e.*, $\rho = 0$).

---

**The speaker recognition user prompts.**
You are an intelligent assistant responsible for speaker recognition.
1. List of Speaker Candidates is as follows:
————————————
`{candidate_str}`
————————————

Note: Among the candidates, 'Others' indicates that the speaker is a role other than the above-listed candidates. This usually means the speaker is a temporary character in the film and television drama (e.g., announcer, police officer, passerby, staff member, etc.) or a main character not present at the scene.
2. Contextual lines have been organized into text format, where each line represents a single line of dialogue with the structure: '[Serial Number] Speaker: Dialogue Line'.
*For example, '[1] Alice: Hello'. If the speaker is marked as 'Unknown', it means the speaker's identity is undetermined for the time being; the label 'Others' has the same meaning as defined above.*
The specific content of the lines is as follows:
————————————
`{json_subtitles}`
————————————

3. The serial number of the target line you need to judge is {j}.

---

**The speaker recognition cheat prompts.**
The actual speaker of the target line is ⟨`the_true_role`⟩. Please provide a reasonable tool calling process and analytical reasoning based on the known answer, and finally present the conclusion. Note that you must pretend to be unaware of the answer when generating the output, and do not reveal the fact that you know the answer in any way.

*Figure 11.* **Top:** the user prompt to provide basic scene information for identifying the speaker of the target line in film and television drama scenes. **Bottom:** the user prompt to guide the generation of a reasonable tool calling and analysis process with the known actual speaker of the target line.

| Duration | LP | DramaSR-LRM | | | | | | | | | |
|---|---|---|---|---|---|---|---|---|---|---|---|
| | | 0.02 | 0.04 | 0.06 | 0.08 | 0.10 | 0.12 | 0.14 | 0.16 | 0.18 | 0.20 |
| *Long (>2s)* | 85.34 | 86.55 | 87.21 | 87.50 | 87.60 | **87.62** | 87.60 | 87.59 | 87.56 | 87.52 | 87.51 |
| *Medium (1s–2s)* | 87.12 | 88.23 | 88.79 | 88.98 | **89.00** | 88.92 | 88.82 | 88.66 | 88.54 | 88.39 | 88.27 |
| *Short (0.5s–1s)* | 82.37 | 84.20 | 85.09 | 85.60 | **85.72** | 85.70 | 85.52 | 85.24 | 85.01 | 84.81 | 84.63 |
| *Very Short (0s–0.5s)* | 67.45 | 70.76 | 73.29 | 74.36 | 75.76 | 76.65 | 76.92 | **77.07** | 76.95 | 76.68 | 76.59 |
| **Total** | 85.49 | 86.78 | 87.45 | 87.74 | **87.82** | 87.79 | 87.71 | 87.59 | 87.48 | 87.36 | 87.27 |

*Table 6.* Speaker recognition accuracy (%) in the subsets of different utterance durations with respect to different confidence sampling thresholds. The dramas are sorted in the same order as in Table 4. **LP**: label propagation (*i.e.*, $\rho = 0$).

### B.1. Full Numerical Results

Table 5 shows drama-wise speaker recognition accuracy with respect to different confidence sampling thresholds. Table 6 shows speaker recognition accuracy in the subsets of different durations with respect to different confidence sampling thresholds.

### B.2. Downstream Video Understanding

Figure 12 shows an example how speaker recognition impacts video captioning and question-answering. Upon the DramaSR-532K benchmark, we will establish a test set comprising thousands of QA pairs to quantitatively evaluate the benefit of speaker reocgnition to video understanding.

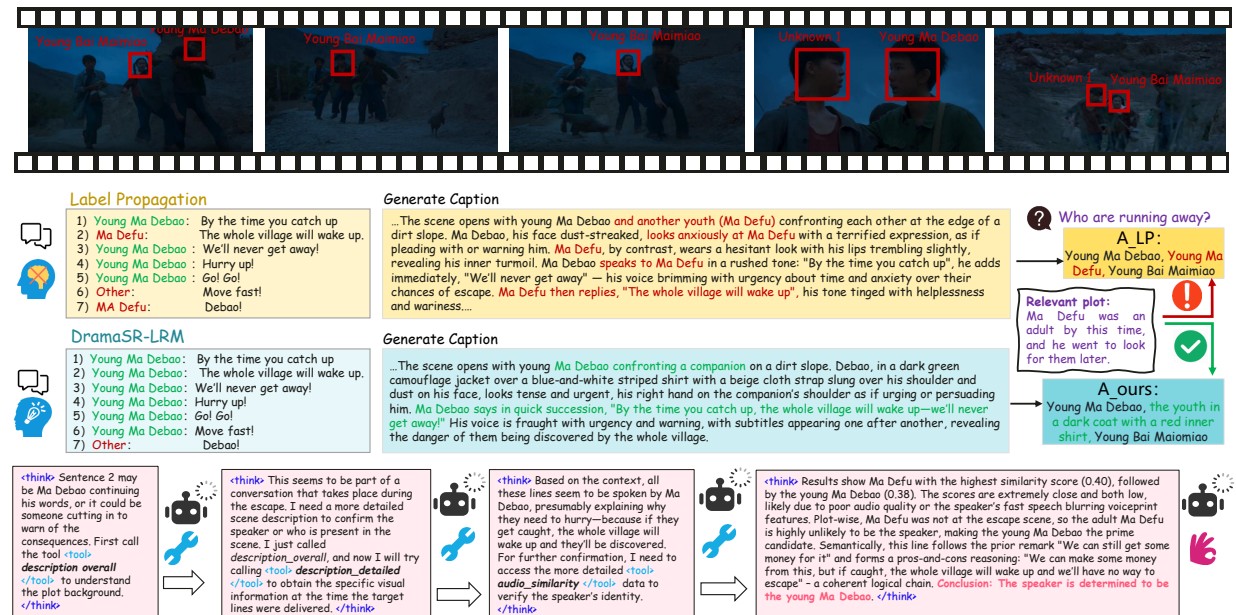

*Figure 12.* An example showing that improved speaker recognition helps downstream video understanding. **Top:** raw video data. **Middle:** speaker recognition results (label propagation baseline and DramaSR-LRM) and the corresponding video captioning and question-answering results, where correct and incorrect outputs are marked in green and red, respectively. **Bottom:** the chain-of-thought produced by DramaSR-LRM during the inference of the second sentence ('*The whole village will wake up*'), where different tool-uses have been highlighted.

