# OpenReview forum: "Reasoning LLM Improves Speaker Recognition in Long-form TV Dramas"
_ICML.cc/2026/Conference — ICML 2026 regular_

### Official Review · Reviewer_jeeX · 2026-03-10

**Soundness:** 2
**Presentation:** 2
**Significance:** 2
**Originality:** 3
**Overall Recommendation:** 2
**Confidence:** 4

**Summary:**

This paper addresses speaker recognition in long-form TV dramas.
In contrast to traditional speaker verification, this work frames a classification task in which each utterance is assigned to a specific character from a candidate set, relying on dialogue context and visual cues beyond acoustic signals alone.
The authors make two contributions. First, they introduce DramaSR-532K, a large-scale benchmark across 13 TV dramas with over 900 characters. Second, they propose DramaSR-LRM, a reasoning model that operates on top of a label propagation baseline and refines difficult cases by invoking pre-computed tools. The model is trained via distillation of reasoning trajectories from a stronger model followed by reinforcement learning. Experiments on 11 held-out dramas show that DramaSR-LRM brings certain improvements over the label propagation baseline, with gains particularly pronounced on short utterances where acoustic features are unreliable.

**Compliance With Llm Reviewing Policy:**

Affirmed.

**Final Justification:**

I appreciate the additional evidence provided in the rebuttal, I raise the Soundness score. I maintain the overall score, as our discussion has revealed several issues that may require substantial revision, including but not limited to:
1. The model relies on an external rule to bypass 80% of samples, as it cannot make confidence judgments itself. The authors attribute this to training data bias and acknowledge that a larger balanced dataset could resolve this, but defer to the final version. I believe this revision is meaningful, but it could require significant revision to the training pipeline.
2. The most straightforward motivation (downstream QA) is not part of the current submission. Integrating it (dataset, metrics, analysis) would be a substantial addition. Moreover, the LP→LRM gain on QA is only 1.7%. I'm still concerned about its practical impact.

**Key Questions For Authors:**

1. Can you provide ablation results removing each tool (voice_sim, video_cap, char_relation) individually? This would clarify whether the improvements stem from multimodal fusion or are dominated by a single modality.
2. What are the primary error patterns of the SFT-only model on easy cases?
3. What is the distribution of the three claimed challenge cases (short utterances, environmental complexity, visual absence) in the dataset? This information is important for assessing the benchmark's coverage.
4. Is there any quantitative evidence, beyond Figure 12, that improved speaker recognition leads to measurable downstream gains?

**Limitations:**

yes

**Strengths And Weaknesses:**

Overall, this work demonstrates certain originality. However, several concerns regarding technical soundness and the justification of significance require further clarification.

**Strengths:**

- **Data contribution**.
This benchmark fills a gap for speaker recognition in the specific setting of long-form TV dramas. The scale and multilingual coverage make it a useful resource not only for drama understanding but also for related problems such as multi-genre speaker recognition.

- **Novel problem-method combination**.
Framing speaker recognition as a tool-use reasoning task for an LRM is a novel attempt in this area. Although the individual technologies are largely well-established (e.g., face recognition-based data collection, LRM-based reasoning), the experiments provide early insights into agentic speaker attribution.

**Weaknesses:**

- **Insufficient motivation**.
The paper positions speaker recognition as a "foundational requirement" for long-form video understanding, yet this premise is not adequately justified. One may argue that current multimodal LLMs (e.g., Gemini, GPT-4o) already possess native video comprehension and character reasoning capabilities.
Besides, this benchmark follows a specific speaker classification setting, but lacks proper motivation to clarify why it is framed like this, and which downstream tasks benefit from this particular formulation (instead of other potentially more general formulations).

- **Concerns on the training approach and model capability**.
The SFT-only model underperforms the non-LLM label propagation baseline, raising concerns about the soundness of the SFT stage. Although the full SFT+RL model surpasses the baseline (85.49% -> 86.93%), it builds on top of baseline attributions and contributes only a 1.44% absolute gain. The best result (87.79%) further relies on external rule-based confidence sampling that retains baseline predictions for ~80% of samples. This suggests the model alone has not learned to appropriately weigh multimodal evidence. The authors should provide error breakdowns by difficulty tier and an analysis of when the LRM helps versus hurts, to clarify the model's actual reasoning capability.


- **Claimed challenges not validated**.
Section 3 motivates the benchmark through three challenges — short utterances, environmental complexity, and visual absence — but only utterance duration is actually evaluated (Table 2). The other two are neither quantified in the data statistics nor tested in experiments.

- **Missing ablation for tool contributions**.
The central claim is that the LRM synthesizes multimodal evidence via tool-use, yet no ablation isolates each tool's contribution. Without ablation studies, it remains unclear whether the gains stem from multimodal fusion or primarily from a single modality. Reporting tool invocation frequencies across the test set would also help substantiate this claim.

**Other Concerns:**
- No inter-annotator agreement or stage-wise accuracy audit is reported for the ground-truth labels; this transparency is expected for a benchmark contribution.
- The confidence sampling threshold ρ=0.10 appears selected on the test set (Table 4) without an independent validation set, risking overfitting to the evaluation data.
- No variance across training runs is reported. Given that the overall improvement is modest (2.30%), reporting standard deviations seems important.

---

> ### Author Rebuttal · Authors · 2026-03-31
>
> Thanks for pointing out that our "data contribution fills a gap for SR", and we present "novel problem-method combination". Concerns addresed below. **Welcome further questions!**
>
> ## Weaknesses
>
> **Q1:** Motivation.
> 1. Can Gemini/GPT-4o do SR natively?
> 2. Why framed like classification?
> 3. Downstream gains?
>
> **A1:** Thanks.
> 1. We tried, but no GPT/Gemini model can join image, text, and audio for SR. Even if the models can take three inputs, they cannot compare audio data to a built library, which is a must for finding the exact ID of the speaker. This is why we designed such a tool-based agent, where (i) the face library is added in video captioning (IDs marked with bounding boxes), and (ii) the speaker library is added with audio similarity. Even with these tools, Gemini-3-Pro needs hints to generate high-quality SFT data (see data curation).
> 2. For long TV dramas that contain 30--100+ named and many unnamed characters, open-set classification is a normal setting for SR. As we release our data to the public, other settings (*e.g.*, end-to-end SR) can be easily created.
> 3. In a follow-up project, we collected 20K+ question-answer (QA) pairs (for plot understanding) on these 13 dramas. In the subset of 18,399 QAs (covering 11 dramas in the SR test set), we used Qwen3-VL-32B for multimodal QA, and different SR results (None: no speaker labels; LP: label prop; LRM: our method; GT: ground-truth) are used as auxiliary clues. As shown below, better SR leads to higher QA acc.
>
> |SR|QA Acc(%)|
> |-|-|
> |None|21.6|
> |LP|70.3|
> |LRM|72.0|
> |GT|80.8|
>
> **Q2:** Training & capability.
> 1. Soundness of SFT.
> 2. Relying on external rule; model's ability to weigh evidence?
> 3. Error breakdowns by difficulty tier.
> 4. When does LRM help/hurt?
>
> **A2:** Thanks.
> 1. This is a known fact of training tool-based agents. The main goal of SFT is to teach the LRM to use these tools, but the model easily over-fits the small-scale SFT data (*i.e.*, 10K in our work). RL is vital for the model to transfer.
> 2. Confidence sampling. It is indeed hard for the LRM to judge "what is a big enough gap in audio similarity". As a main point, we note that the value of LRM lies in judging hard cases; we encourage it to judge more cases, and the cost is a higher risk of making errors in easy cases. Fortunately, this can be fixed by a simple rule (and it also speeds up inference).
> 3. Breakdown. Please see Table 5 in the Appendix. The trend is similar in all tiers.
> 4. The LRM helps to integrate multimodal clues to solve corner cases, and hurts mostly when it makes errors in easy cases.
>
> **Q3:** Challenges not validated.
>
> **A3:** Thanks.
> * **Short utterances.** Ratio: Long=34.1%, Medium=55.2%, Short=10.0%, VeryShort=0.7%.
> * **Environmental complexity.** We group the utterances by # candidates. LRM is better in all sets.
>
> |#Cand|Ratio(%)|LP(%)|LRM(%)|
> |-|-|-|-|
> |1--2|40.6|84.91|87.98|
> |3--4|40.8|86.12|87.94|
> |5+|18.6|85.46|86.07|
>
> * **Visual absence.** There are ~9.6K (1.8%) off-screen utterances, *i.e.*, GT speaker's face not appearing in ±30s context. The model must assign an `[OTHERS]` label, after which an extra step is done to assign the correct speaker label. LP and LRM succeed in ~13.4% and ~52.4% cases, respectively: LP mostly predicts an active character, and LRM does much better via reasoning.
>
> **Q4:** Tool ablation and invocation.
>
> **A4:** Thanks! Results are shown below. Conclusions: (i) `audio_sim` is the most useful tool, without which the accuracy drops a lot. (ii) `video_cap` also helps, in particular when captions include speaking actions. (iii) `char_relation` brings a small gain, but it offers useful clues for hard cases, mostly very short relation-calls (*e.g.*, "Dad" or "Mom").
>
> |Tool-set|Acc(%)|Drop(%)|
> |-|-|-|
> |All|87.79|--|
> |w/o `audio_sim`|72.61|15.18|
> |w/o `video_cap`|86.76|1.03|
> |w/o `char_relation`|87.55|0.24|
>
> |Tool|`audio_sim`|`video_cap`|`char_relation`|
> |-|-|-|-|
> |Freq(%)|100.0|33.1|4.6|
>
> ## Other Concerns
>
> **Q5:** Inter-annotator agreement; stage-wise audit.
>
> **A5:** There are only two stages of labeling, Stage 1 for initial labeling and Stage 2 for double-check. Stage 2 fixes about 1.4% labels in Stage 1, but many were caused by carelessness of Labeler 1. The agreement between labelers is about 99.6%.
>
> **Q6:** Threshold ρ=0.10 on validation.
>
> **A6:** Val set has 1048 samples. Hard-case sampling was used (see Sec 4.4), so accuracy looks lower. ρ=0.10 is best. We will revise our description of choosing ρ.
>
> |ρ|LP|0.02|...|0.08|0.10|0.12|...|0.20|
> |-|-|-|-|-|-|-|-|-|
> |Acc(%)|77.77|78.91|...|80.15|80.25|80.06|...|79.67|
>
> **Q7:** Variance across runs.
>
> **A7:** Standard deviation (of total acc.) across 5 runs is 0.39%, much smaller than the gain, 2.30%.
>
> ## Key Questions
>
> **Q8:** Error patterns of SFT-only.
>
> **A8:** Over-fitted on training data, the model tends to fully (and sometimes repeatedly) call tools until reaching max rounds and running into failure. RL fixes the bad habit with a negative reward to failure and number of tool calls.

---

> > ### Author Rebuttal · Reviewer_jeeX · 2026-04-03
> >
> > Thanks for the detailed clarification and additional experiments. Concerns regarding inter-annotator agreement (Q5), validation set for ρ (Q6), variance across runs (Q7), and SFT error patterns (Q8) are adequately addressed. I will adjust the Soundness score. However, some important concerns about motivation and core model capability remain unresolved, as detailed below.
> >
> > **Q1 & Q8**
> > - I agree that downstream QA benefit is a strong motivation, and I believe a paper that integrates this follow-up QA project would constitute a more complete contribution.
> > - Even so, the LP→LRM gain on downstream QA is only 1.7% (without std reported), which makes it hard to argue that the proposed method meaningfully addresses the practical problem as motivated.
> > - Besides, the assertion that SR is "a foundational requirement" for long-form video understanding remains insufficiently justified. While current MLLMs may not support audio-library well, it is still unclear why SR must exist as an independent module, rather than being implicitly learned end-to-end.
> >
> > Therefore, I maintain my concern about the motivation and significance of this submission.
> >
> > **Q2**
> > - While SFT overfitting on small data is common, the behavior described in A8 (repeatedly calling tools until max rounds and failing) goes beyond overfitting — it warrants more discussion in the paper.
> > - I maintain my concern about model capability. Rather than relying on an external rule to bypass 80% of samples, it would be more convincing to discuss why the model cannot learn this confidence judgment itself.
> > - Table 5 reports accuracy across tiers, not the error analysis. A flip analysis (wrong→right vs. right→wrong among LRM-intervened samples) is needed to distinguish whether the net gain reflects genuine reasoning or near-random flipping. The qualitative answer in A4 does not resolve this.
> >
> > **Q3**
> >
> > I appreciate the effort. The visual absence result resolves my concern on that front. However, the environmental complexity results show diminishing gains as candidate count increases (3.07% for 1–2 vs. 0.61% for 5+), which contradicts the claim that LRM's strength is disambiguating in crowded scenes.
> >
> > **Q4**
> >
> > I appreciate the effort. The results confirm that audio_sim dominates, while video_cap and char_relation contribute only 1.03% and 0.24% respectively. This suggests the paper's central narrative of 'multimodality may be overstated.
> >
> > --
> >
> > In light of the rebuttal, I raise Soundness to reflect the newly provided ablation, variance, and validation results. I maintain the overall score, as the unresolved concerns — limited motivation, marginal multimodal contribution (Q4), and insufficient evidence for the claimed reasoning capability (Q2) — affect the core claims of this work.

---

> > > ### Author Response · Authors · 2026-04-03
> > >
> > > Thank you for the detailed feedback. We truly appreciate your efforts in helping us improve this work and are glad that several of your previous concerns have been resolved. Regarding your remaining points, thanks for this extra opportunity for discussion, and we hope our responses below address them fully. **Of course, further discussions are welcome!**
> > >
> > > **Q1:** Motivation and significance.
> > > 1. QA benefit is strong.
> > > 2. LP→LRM gain on QA is only 1.7% (no std), hard to argue.
> > > 3. "SR is a *foundational requirement* for ...": Why SR must exist independently?
> > >
> > > **A1:**
> > > 1. We appreciate your recognition of the QA downstream task and will feature it prominently in the Experiments section.
> > > 2. We tested the 5 SR results (SR accuracy's std-dev: 0.39%; see **Q7** of the last rebuttal) on the downstream QA tests, and the QA accuracy's std-dev is **0.33%**, smaller than 1.7%, implying that the gain is statistically meaningful. By the way, 1.7% is **not** small as it seems, especially when LP has achieved a high QA accuracy of 70.3%. Note: an increase from 70.3% to 72.0% means to **reduce the error rate by 8.4%**.
> > > 3. We understand your concern and will tone down our claim to state that "SR offers *useful information* for long-form video understanding." Regarding why an **explicit** SR module is necessary: A long-form drama contains 100s of characters and 1000s of dialogue sections. Expecting an end-to-end MLLM to maintain such a massive, complex scene-character memory is currently unrealistic. While end-to-end capabilities will undoubtedly evolve, an explicit memory (with tool-based inference) yields far more reliable identity recognition today. Exploring these explicit mechanisms offers valuable design insights for training the powerful MLLMs of the future. **Our research aligns with this objective and is a stepping stone to your ambitious perspective.**
> > >
> > > **Q2:**
> > > 1. The behavior (repeatedly calling tools) warrants more discussion.
> > > 2. Model capability: why cannot the model learn judgment itself?
> > > 3. A flip analysis is needed.
> > >
> > > **A2:** Thanks. These behaviors trace back to our training data sampling strategy. To prevent the model from learning a trivial, conservative strategy (*i.e.*, always choosing the highest audio similarity), we used hard-case sampling (Sec 4.4) to filter out most easy samples. Trained on this highly difficult, biased distribution, the model maximizes its reward by learning to **over-intervene**, which necessitates post-training confidence calibration. While curating a vastly larger, balanced SFT/RL dataset would theoretically allow the model to learn this judgment natively (we will report results in the final version), our calibration rule is a practical and effective solution for the current scope.
> > > 1. Because the SFT data is heavily biased toward complex cases requiring multiple tool-calls, the model defaults to exhaustive search, unaware that repeated calls yield no new information. After RL, such behavior is penalized and vanishes. We will expand on this dynamic in the text.
> > > 2. See the discussions above. From another viewpoint, we forced the model to out of the local optimum that relies only on audio similarity (RL can quickly learn a simple behavior to get easy rewards), and used calibration afterward.
> > > 3. Now we fully understand what you meant. Below we show the breakdown (with ρ=0.1) in different tiers. For all of them, the intervention precision exceeds 64%, implying **the net gain is driven by genuine reasoning, not random flipping**.
> > >
> > > |Set|LRM(%)|=LP+Help-Hurt(%)|Prec(%)=Help/(Help+Hurt)|
> > > |-|-|-|-|
> > > |ALL|87.79|=85.49+4.86-2.56|65.5|
> > > |Long|87.62|=85.34+4.48-2.20|67.0|
> > > |Medium|88.92|=87.12+3.98-2.11|64.6|
> > > |Short|85.70|=82.37+7.09-3.76|65.3|
> > > |VeryShort|76.65|=67.45+16.63-7.43|69.1|
> > >
> > >
> > > **Q3:** Environmental complexity.
> > >
> > > **A3:** To clarify, our claim is not that "LRM shows *larger* gains as candidate count increases", but rather that "LRM achieves *consistent* gains across various environmental complexities". In crowded scenes, distinguishing speakers inherently relies more heavily on core biometrics. While the absolute gain shrinks, other modalities still provide useful information to surpass the LP baseline.
> > >
> > > **Q4:** Multimodality may be overstated.
> > >
> > > **A4:** We respectfully argue that 1.03%/0.24% gains are not small, contributing **44.8%/10.4%** of the overall 2.30% gain, which is significant given a high LP baseline, 85.49%. To show that other modalities help more **when audio features are less stable**, we report ablation results on different tiers below. One can see the increasing need of `video_cap` and `char_relation` in shorter utterances.
> > >
> > > |Set|LRM(%)|w/o `audio_sim`(%)|w/o `video_cap`(%)|w/o `char_relation`(%)|
> > > |-|-|-|-|-|
> > > |ALL|87.79|72.61(-15.18)|86.76(-1.03)|87.55(-0.24)|
> > > |Long|87.62|72.48(-16.14)|86.45(-1.17)|87.41(-0.21)|
> > > |Medium|88.92|73.21(-15.71)|87.81(-1.11)|88.68(-0.24)|
> > > |Short|85.70|70.32(-15.38)|83.30(-2.40)|85.33(-0.37)|
> > > |VeryShort|76.65|62.91(-13.74)|72.33(-4.32)|75.66(-0.99)|

---

### Official Review · Reviewer_7NEq · 2026-03-12

**Soundness:** 3
**Presentation:** 2
**Significance:** 4
**Originality:** 3
**Overall Recommendation:** 5
**Confidence:** 3

**Summary:**

This paper addresses the challenge of speaker recognition in long-form TV dramas by introducing DramaSR-532K, a large-scale benchmark comprising 532K annotated utterances across 13 series. To overcome the limitations of purely acoustic methods on short utterances, the authors propose DramaSR-LRM. This method employs a large reasoning model coupled with multimodal tool-use  to iteratively refine initial speaker labels. Trained via supervised fine-tuning and GRPO-style reinforcement learning, the model delivers a 2.30% absolute accuracy gain over a strong label-propagation baseline.

**Compliance With Llm Reviewing Policy:**

Affirmed.

**Final Justification:**

The authors address my main concerns.

**Key Questions For Authors:**

please see the weaknesses

**Limitations:**

yes

**Strengths And Weaknesses:**

Strengths
- The reasoning-first approach effectively synthesizes acoustic, visual, and relational cues via explicitly designed multimodal tool-use, which is a highly appropriate and innovative formulation for identity-dense narratives.
- The implementation of a confidence-based selective reasoning mechanism and an iterative refinement loop is highly practical, ensuring that expensive LLM inference is deployed only for ambiguous cases.
- The DramaSR-532K dataset is a substantial contribution to the community. The extensive cross-drama evaluation convincingly demonstrates the model's value, particularly highlighting its robust performance improvements on short and very short utterances where standard acoustic biometrics fail

Weaknesses
- The current evaluation treats any prediction as valid when the ground truth is [UNKNOWN], which may artificially inflate the reported accuracy. The paper lacks transparency regarding the exact prevalence of these cases in the test set.
- The heavy reliance on OCR-extracted subtitles and a ±30s face-based visual neighborhood assumption embeds strict priors that may bias the candidate pool and systematically underplay off-screen or occluded speech.
- The experimental section lacks an ablation study isolating the individual contributions of each tool (voice_sim, video_cap, char_relation) and misses an "oracle tool" analysis to bound performance headroom. Additionally, the paper omits empirical comparisons with recent, strong AV-LLM approaches optimized for speaker attribution (e.g., D-ORCA).
- Despite the confidence sampling optimizations, the reported amortized inference cost of 3 GPU-seconds per utterance remains computationally expensive for processing massive, real-world TV corpora.

---

> ### Author Rebuttal · Authors · 2026-03-31
>
> We thank the reviewer for recognizing our problem formulation, algorithm design, and the value of our dataset to the community. We address the reviewer's concerns below. We welcome the reviewer for further discussions.
>
> ## Weaknesses
>
> **Q1:** The current evaluation treats any prediction as valid when the ground truth is `[UNKNOWN]`, which may artificially inflate the reported accuracy. The paper lacks transparency regarding the exact prevalence of these cases in the test set.
>
> **A1:** Thanks for the question. The `[UNKNOWN]` tags are very rare, appearing in only <100 (out of 532K) utterances (<0.02%). Excluding these from the evaluation reduces both the label propagation (LP) and LRM overall accuracy by 0.002%. Our LRM remains 2.3% higher than the LP baseline, keeping all conclusions unchanged.
>
> **Q2:** The heavy reliance on OCR-extracted subtitles and a ±30s face-based visual neighborhood assumption embeds strict priors that may bias the candidate pool and systematically underplay off-screen or occluded speech.
>
> **A2:** We agree.
> * Bypassing OCR subtitles (using raw audio) is a harder setting requiring speaker segmentation. We focused on recognition here but will investigate end-to-end settings in future work. Upon releasing our data, the community can easily explore this.
> * Long-form dramas feature massive casts (30--100+ characters), and some voice actors may play multiple roles. Visual anchors are practical currently. In the future research, we plan to bypass the ±30s constraint by combining face and speaker recognition via an online active ID registration algorithm.
> * Our algorithm **can** recognize off-screen speakers. It initially assigns `[OTHERS]` (because the true speaker is not in the original candidate list). If needed, it extends the candidate list with 3–5 characters sharing similar voiceprints and reasons again. In our experiments, it correctly recognizes 52.4% of these cases, significantly outperforming the LP baseline which succeeds in 13.4% of these cases.
>
> **Q3:** The experimental section lacks an ablation study isolating the individual contributions of each tool (`voice_sim`, `video_cap`, `char_relation`) and misses an "oracle tool" analysis to bound performance headroom. Additionally, the paper omits empirical comparisons with recent, strong AV-LLM approaches optimized for speaker attribution (e.g., D-ORCA).
>
> **A3:** Good question! During the rebuttal, we ablated the tools:
>
> | Tool-set | Acc | Drop |
> |---|---|---|
> | Full | 87.79% | -- |
> | w/o `audio_sim` | 72.61% | 15.18% |
> | w/o `video_cap` | 86.76% | 1.03% |
> | w/o `char_relation` | 87.55% | 0.24% |
>
> Key observations:
> * `audio_sim` is the most crucial tool (the only way LRM extracts audio), and removing it causes a dramatic accuracy drop.
> * `video_cap` helps significantly, particularly when descriptions include speaking-related actions.
> * `char_relation` contributes less overall but resolves hard cases, such as very short utterances with calling relationships (e.g., "Dad" or "Mom").
>
> Regarding D-ORCA (arXiv:26**02**.07960), it appeared **after** the ICML submission deadline (Jan 28th), so we were unaware of it. Furthermore, D-ORCA focuses on short clips (about 100s), whereas we focus on long-form dramas (about 40 hours). This requires different designs, primarily our speaker library (unnecessary for short clips) and, consequently, the `audio_sim` tool (D-ORCA uses no tools). We appreciate the reference, learned from their audio incorporation and reward design, and will cite and discuss it in the final version.
>
> **Q4:** Despite the confidence sampling optimizations, the reported amortized inference cost of 3 GPU-seconds per utterance remains computationally expensive for processing massive, real-world TV corpora.
>
> **A4:** Thank you for this kind reminder! We discovered an incorrect vLLM setting. We assumed `max_workers=20` meant 20 threads *per GPU*, but it allocated 20 threads *total* across 8 GPUs. Setting `max_workers=256` speeds up inference 9x, requiring only 0.33 GPU-seconds per utterance. With confidence-sampling, processing a 50K-utterance drama now takes under 6 minutes on an 8-GPU server. (Note: Video captioning takes longer and is our next optimization target.)

---

> > ### Author Rebuttal · Reviewer_7NEq · 2026-04-01
> >
> > Thank you for the responses. My questions are all solved

---

> > > ### Author Response · Authors · 2026-04-04
> > >
> > > Dear Reviewer,
> > >
> > > Thanks for your response. More importantly, thanks for your comment, which helps us improve this work. We will add the content to the final paper.
> > >
> > > Best,
> > >
> > > Authors of ICML 2026 Conference Submission 3280

---

### Official Review · Reviewer_wiKA · 2026-03-14

**Soundness:** 3
**Presentation:** 3
**Significance:** 3
**Originality:** 3
**Overall Recommendation:** 4
**Confidence:** 4

**Summary:**

This paper presents DramaSR-532K, a large-scale benchmark of 532K annotated utterances from 13 long-form TV dramas, designed to study speaker recognition as character attribution in complex, multimodal narrative settings. It also introduces DramaSR-LRM, a large reasoning model that integrates audio similarity, hierarchical video captioning, and dialog-derived character relations to iteratively refine labels produced by a label-propagation baseline. Evaluated on 11 held-out dramas, DramaSR-LRM improves utterance-level accuracy from 85.49% to 87.79%, with particularly strong gains on very short utterances where acoustic cues are unreliable.

**Compliance With Llm Reviewing Policy:**

Affirmed.

**Key Questions For Authors:**

1. How many ground-truth [UNKNOWN] and multi-speaker utterances exist in the test sets, and how do reported accuracies change when (a) excluding UNKNOWN lines and (b) requiring exact-match for multi-speaker lines?
2. Can you provide inter-annotator agreement, percent of pseudo-labels corrected by humans, and an estimate of final label noise after quality control?
3. What is the convergence/stopping rule in the iterative loop, and do you observe oscillations or drift? Any theoretical or empirical guarantees?

**Limitations:**

yes

**Strengths And Weaknesses:**

Strengths:
1. The formulation of speaker recognition as multimodal character attribution with explicit tool-use (voice_sim, video_cap, char_relation) in a reasoning loop is a fresh and compelling approach for long-form narratives.
2. Iterative refinement that re-computes dynamic tools based on updated labels is well-motivated and shows self-improvement behavior.
3. A dataset of this scale and with character-centric labels is valuable for the community; it enables research on long-form, high-character-density video understanding.
Weaknesses:
1. The evaluation protocol counts any prediction as correct on [UNKNOWN] ground-truth lines and accepts any of multiple speakers for multi-speaker lines; this may inflate accuracy and obscure differences between methods.
2. The neighborhood assumption (±30s face presence) restricts candidate sets for both baseline and evaluation, possibly biasing the task in favor of visually anchored methods and under-representing off-screen speech difficulty.
3. Reliance on large proprietary models for tool generation (e.g., Gemini teacher, Qwen3-VL-32B captioning) raises reproducibility concerns and potential leakage of prior knowledge about popular series.

---

> ### Author Rebuttal · Authors · 2026-03-31
>
> We thank the reviewer for the recognition of our problem formulation, algorithm design, and the value of our dataset to the community. We address the concerns below. We welcome the reviewer for further discussions.
>
> ## Weaknesses
>
> **Q1:** The evaluation protocol counts any prediction as correct on `[UNKNOWN]` ground-truth lines and accepts any of multiple speakers for multi-speaker lines; this may inflate accuracy and obscure differences between methods.
>
> **A1:** We discuss the two cases separately.
> * The `[UNKNOWN]` tag appears rarely, occupying <100 out of 532K (<0.02%) utterances. Excluding these from evaluation reduces both label propagation (LP) and LRM accuracy by only 0.002%. LRM still outperforms LP by 2.3\%.
> * Multi-speaker lines occupy ~2K out of 532K (<0.4%) utterances. Since current methods (LP and LRM) treat each line as an inseparable unit, requiring an exact match causes all methods to fail, reducing overall accuracy by 0.34% and 0.35%, respectively. LRM still outperforms LP by 2.3\%. We agree that distinguishing multiple speakers is important. In the future, we will train the LRM to detect multi-speaker lines and use an audio separation tool for voiceprint extraction. We can easily build training data by merging adjacent single-speaker lines.
>
> **Q2:** The neighborhood assumption (±30s face presence) restricts candidate sets for both baseline and evaluation, possibly biasing the task in favor of visually anchored methods and under-representing off-screen speech difficulty.
>
> **A2:** We agree.
> * A long-form drama has many (30--100+) characters, and a voice actor may perform multiple characters. Using visual anchors is practical for now.
> * Our algorithm **can** recognize off-screen speakers. It first assigns `[OTHERS]`. If necessary, it extends the candidate list by adding 3 to 5 characters with the most similar voiceprints and performs another reasoning step. In experiments, 52.4% of off-screen cases are correctly recognized.
> * In the future, we will bypass the hard constraint by combining face and speaker recognition with an online registration algorithm for active IDs.
>
> **Q3:** Reliance on large proprietary models for tool generation (*e.g.*, Gemini teacher, Qwen3-VL-32B captioning) raises reproducibility concerns and potential leakage of prior knowledge about popular series.
>
> **A3:** Thanks for the question.
> * Reproducibility: We generated SFT data 3 times (2 times with GPT-5, 1 time with Gemini-3-Pro), reporting similar accuracy. SFT gives basic tool-call abilities, but RL (not relying on proprietary models) is more important for final accuracy. **We will release the SFT data to guarantee reproducibility.** By the way, Qwen3-VL models are open-sourced.
> * Information leakage: The proprietary models and our base model (Qwen3-8B) may contain prior knowledge, but none work well enough directly. As described in Section 4.4, data curation with hints is crucial. This shows the importance of our pipeline (SFT followed by RL) to unlock the potential of LLMs.
>
> ## Detailed Questions
>
> **Q4:** How many ground-truth `[UNKNOWN]` and multi-speaker utterances exist in the test sets, and how do reported accuracies change when (a) excluding `[UNKNOWN]` lines and (b) requiring exact-match for multi-speaker lines?
>
> **A4:** Please see **Q1**.
>
> **Q5:** Can you provide inter-annotator agreement, percent of pseudo-labels corrected by humans, and an estimate of final label noise after quality control?
>
> **A5:** The average inter-annotator agreement is about 99.6%. The percent of pseudo-labels corrected by humans is 1 minus the LP accuracy (about 14.5%). By estimation, the final label noise is about 0.5%.
>
> **Q6:** What is the convergence/stopping rule in the iterative loop, and do you observe oscillations or drift? Any theoretical or empirical guarantees?
>
> **A6:** We iterate until less than 0.1% of predicted speaker labels change in a single round. This condition is always satisfied after 2--4 rounds, implying very few labels oscillate. While not theoretically guaranteed, a label is mostly determined by its own acoustic biometric and captions (unchanged throughout iterations) and impacted by its context labels. Once the context labels get stable, changes become fewer in later iterations.

---

> > ### Author Rebuttal · Reviewer_wiKA · 2026-04-04
> >
> > My concerns have all been resolved.

---

> > > ### Author Response · Authors · 2026-04-04
> > >
> > > Dear Reviewer,
> > >
> > > Thanks for your response. We believe that your comment helps us improve this work. We will add the content to the final paper. **We would appreciate it if you may consider adjusting your score since the previous concerns have all been resolved.**
> > >
> > > Best,
> > >
> > > Authors of ICML 2026 Conference Submission 3280

---

### Decision · Program_Chairs · 2026-04-30

**Decision:**

Accept (regular)

**Comment:**

This paper introduces DramaSR-532K, a large-scale benchmark for speaker recognition in long-form TV dramas, alongside DramaSR-LRM, a multimodal reasoning model that leverages tool-use to iteratively attribute utterances to characters. The reviewers commended the scale and practical value of the dataset, as well as the fresh formulation of speaker recognition as a multimodal reasoning task, noting its particular effectiveness on short utterances where traditional acoustic cues often fail. During the review period, reviewers raised several concerns, including the need for a detailed ablation of the individual tools, questions regarding the computational overhead, potential biases in the evaluation protocol, and the broader motivation for an independent speaker recognition module. In a rebuttal, the authors addressed the majority of these issues by providing tool ablations, correcting a vLLM configuration to show reduced inference costs, and presenting an analysis and downstream QA evaluation to solidify the method's practical impact. Although one reviewer maintained reservations regarding the model's reliance on an external confidence threshold and its foundational motivation, the broader consensus acknowledges the substantial value of the benchmark and the originality of the agentic approach. Given that the rebuttal robustly resolved the primary technical and empirical concerns for the majority of the reviewers, this paper represents a solid, innovative contribution to the community and therefore is recommended for acceptance.